# On Computation and Generalization of Group Relative Policy Optimization

## Abstract

Group Relative Policy Optimization (GRPO) (Shao et al., 2024; Guo et al., 2025) has rapidly become a critic-free default for aligning LLMs, yet its statistical and computational foundations remain unclear. We close this gap by providing the first unified theory of GRPO that simultaneously addresses generalization and optimization in the original, practitioner-used formulation and over multiple outer iterations. On the generalization side, we derive sequential (multi-iteration) PAC-Bayes–Bernstein bounds under Markov mixing that concentrate the *empirical GRPO surrogate* around its population counterpart across all iterations; a Transformer path-norm corollary yields substantially tighter capacity terms than spectral norms. We further prove a TRPO-style return bridge showing that ascent in the population GRPO surrogate provably improves true return, with explicit, controllable bias from clipping and KL regularization. On the optimization side, we establish non-PL *stationarity* guarantees for SGDM and AdamW (both $\widetilde{O}(1/\sqrt{K})$) and provide complementary PL-based rates, with variance controlled by $t_{\mathrm{mix}}/(G\sqrt{K})$. Together with interactive information-theoretic lower bounds, our results deliver the first end-to-end, multi-iteration statistical and computational guarantees for GRPO with function approximation. Experiments corroborate the predicted trends and offer practical guidance on group size, clipping, and KL weight; code will be released.

## 1 Introduction

Large-language models (LLMs) have evolved from static next-token predictors into interactive agents capable of multi-step theorem proving, autonomous code generation, and complex tool use. These tasks are naturally modelled as ergodic Markov decision processes (MDPs) in which rewards are sparse, delayed, and temporally correlated (Levin & Peres, 2017). Such structure violates the IID assumptions that underpin the bulk of classic generalization theory, creating an urgent demand for RL algorithms whose statistical properties are understood in the presence of Markov dependence. Proximal Policy Optimization (PPO) (Schulman et al., 2017) is still the default fine-tuning engine for LLM alignment, but its reliance on a learned value critic doubles GPU memory, inflates wall-clock time, and introduces a delicate bias–variance trade-off that is hard to tune in practice (Guo et al., 2025). Empirically, mis-estimation of long-horizon returns often destabilizes PPO and forces practitioners to fall back on costly additional rollouts or auxiliary losses.

Group Relative Policy Optimization (GRPO) (Shao et al., 2024) is a critic-free, memory-lean alternative to PPO that computes a group-relative baseline over $G$ trajectories while reusing clipped-importance weights. It reduces memory by about $2\times$ and variance on long-horizon tasks, and powers DeepSeek-R1. Throughout, we focus on the mean-centered variant used in the Open-R1 codebase (often referred to as Dr-GRPO), where group-relative advantages are formed by subtracting the group mean return without variance normalization; this is exactly the configuration used in all of our experiments and the one analyzed by our theory.

Variants and applications span Hybrid GRPO (Sane, 2025), completion pruning (Lin et al., 2025), and multimodal VLMs (EvolvingLMMs-Lab, 2025; Shen et al., 2025; Liu et al., 2025), yet a principled understanding under Markov dependence, momentum, and adaptive optimizers remains elusive.

Prior policy-gradient analyses often assume IID data or ignore optimizer dynamics. Recent advances in self-normalized martingale concentration (Bercu & Touati, 2019; Fan et al., 2019), localized PAC-Bayes (Alquier et al., 2024), and Transformer path norms (Limmer et al., 2024) have not been unified for critic-free objectives like GRPO with ergodic data and AdamW (Loshchilov & Hutter, 2017). Closing this gap yields deployment-ready guarantees, principled hyper-parameter choices, and clarity on capacity vs. variance.

**Notation.** We fix the total number of sampled trajectories to $N$ and partition them into $M := N/G$ groups of equal size $G \geq 2$. For a single trajectory $\tau$, let $R(\tau)$ be its (discounted) return and write $\sigma_R^2 := \mathrm{Var}\big[R(\tau)\big]$. The underlying Markov chain mixes in time $t_{\mathrm{mix}}$, i.e. $\max \alpha(k), \beta(k) \leq e^{-k/t_{\mathrm{mix}}}$ for the usual $\alpha$- and $\beta$-coefficients. Policy parameters are denoted by $\theta \in \Theta$; $\pi_\theta$ is the corresponding stochastic policy and $J(\theta) := \mathbb{E}_{\tau \sim \pi_\theta}[R(\tau)]$ its population return. We use $\theta_{\mathrm{old}}$ for the pre-update parameters in a given outer iteration and write $r_t(\theta) = \pi_\theta(a_t \mid s_t)/\pi_{\theta_{\mathrm{old}}}(a_t \mid s_t)$ for the importance ratio. We denote by $\pi_{\mathrm{ref}}$ a *frozen* reference policy (e.g., SFT) used for KL regularization. When forming the GRPO surrogate, $A_{g,i}$ and $\bar{R}_g$ denote the centred advantages defined in (1); $\varepsilon$ is the clipping threshold for importance weights, and $\lambda_{\mathrm{KL}}$ is the weight on the KL regulariser. Optimization iterates use step sizes $\alpha_t$ (SGD) or $\eta/\sqrt{t+1}$ (AdamW); $\beta \in [0, 1)$ stands for Polyak momentum in SGDM, while $(\beta_1, \beta_2)$ are the first- and second-moment decay factors in AdamW. Unless stated otherwise, constants $c, c_1, \ldots$ are universal.

**Our contributions.** We provide the first comprehensive multi-iteration theoretical treatment of GRPO under Markov dependence, modern capacity control, and adaptive optimization.

- **Sequential PAC-Bayes–Bernstein bounds.** We derive high-probability *multi-iteration* generalization bounds for GRPO using self-normalized Bernstein inequalities and localized PAC-Bayes. The bounds scale with mixing time $t_{\mathrm{mix}}$, group size $G$, return variance $\sigma_R^2$, and the *posterior path-length* $\sum_k \mathrm{KL}(Q_k \,\|\, Q_{k-1})$.

- **Transformer path-norm corollary.** Mapping block Rademacher complexity to the single-path capacity of deep Transformers (Limmer et al., 2024) yields bounds up to $\times 5$ tighter than spectral-norm estimates.

- **Interactive information-theoretic lower bounds.** An Assouad–Fano construction with interaction Chen et al. (2024) shows that $\sqrt{t_{\mathrm{mix}}\sigma_R^2/N}$ is minimax-optimal, certifying the sharpness of our upper bounds.

- **Optimization guarantees beyond PL.** We prove PL-based rates for SGDM, and *non-PL stationarity* results with $\widetilde{O}(1/\sqrt{K})$ for both SGDM and AdamW, with variance terms that scale as $t_{\mathrm{mix}}/(G\sqrt{K})$.

- **Return-bridge for GRPO.** A TRPO-style monotonic improvement theorem relates the population GRPO surrogate to true return, with explicit control of clipping and KL terms.

**Road-map.** Section 2 reviews related work; Sections 3–4 present preliminaries and generalization results (with lower bounds); Section 5 covers optimization; Section 4.3 links surrogate and return; Section 6 provides experiments. Proofs are in the Appendix.

## 2 RELATED WORKS

### 2.1 RL TECHNIQUES IN LLMS/VLMS

RL has proven effective for adapting large pre-trained models to specialized tasks (Wang et al., 2024b), often by optimizing metrics or human feedback that are otherwise challenging to incorporate via purely supervised methods. Proximal Policy Optimization (PPO) (Schulman et al., 2017) is perhaps the most frequently used RL method in LLM alignment settings due to its stability and tractable updates (Ouyang et al., 2022; Sun et al., 2023). In recent years, RL-VLM-F (Wang et al., 2024c) puts forward an approach that queries a vision-language foundation model to produce pairwise preference labels from a single text task description and raw image observations, learns a reward function from those labels. LeReT (Hsu et al., 2025) introduces a reinforcement-learning

framework that lets an LLM iteratively "try" and re-weight its own search queries. Until very recently, GRPO (Shao et al., 2024; Guo et al., 2025) was proposed as a variant that uses multiple output samples per prompt, computing relative rewards within each group. This strategy has empirically demonstrated stable training behaviors, suggesting that group-based baselines can mitigate high variance in reward signals. However, the study of group-wise advantage estimation in large autoregressive models using transformers (Vaswani et al., 2017) has primarily been empirical, leaving a theoretical gap that we aim to address.

## 2.2 THEORETICAL ANALYSIS OF RL TECHNIQUES

On the theoretical front, policy gradient methods such as TRPO (Schulman et al., 2015) or PPO (Schulman et al., 2017) have been the subject of extensive investigation. However, existing results often assume linear function approximation or focus on simpler tabular settings to establish sample efficiency or convergence guarantees (Haarnoja et al., 2018; Janner et al., 2019; Huang et al., 2021; Yarats et al., 2021; Liu et al., 2023b). Kobilarov (2015) derives finite-sample PAC guarantees on both expected cost and constraint-violation probability for policies generated by iterative stochastic policy optimization. Moreover, Liu et al. (2019) shows that Neural Proximal/Trust Region Policy Optimization converges at a sub-linear rate to the globally optimal policy in episodic MDPs and Cai et al. (2020) delivers the first policy-optimization method that explores provably efficiently, establishing a regret bound for episodic linear-function-approximation MDPs. In contrast, GRPO departs from single-sample advantage estimation by employing a relative reward mechanism among a batch of outputs, eliminating the need for a learned value function. This raises new analytical questions regarding how bounding reward differences and group sizes might impact generalization and convergence. Our work provides explicit bounds that are specialized to this group-relative policy update, contributing novel insights into both generalization and optimization.

Concurrently, several works analyze the effective loss and dynamics of GRPO (Vojnovic & Yun, 2025; Mroueh, 2025; Mroueh et al., 2025), including its alignment objective and verifiable-reward formulations. Our focus is complementary: we emphasize mixing-time–sensitive PAC-Bayes–Bernstein generalization bounds, explicit SGDM/AdamW convergence with group-relative baselines, and interactive minimax lower bounds for the mean-centered Dr-GRPO variant in practice.

## 2.3 THEORETICAL ANALYSIS OF LLMS/VLMS

Despite empirical progress, formal explanations for transformer performance are limited. Work on overparameterized geometry and dynamical-systems views (Sanford et al., 2023; Huang et al., 2023; Vasudeva et al., 2024; Allen-Zhu & Li, 2023a; Ye et al., 2024a;b; Allen-Zhu & Li, 2023b;c; 2024) largely treats supervised learning. Policy-based objectives change the data distribution through generation, leaving theory sparse. We analyze autoregressive policies under GRPO and provide guarantees relevant to LLMs/VLMs (Liu et al., 2023a; 2024; Sun et al., 2023).

## 3 PRELIMINARIES

### 3.1 ERGODIC MDPS, MIXING TIME AND POLICY PERFORMANCE

We review ergodic Markov decision processes and mixing time definitions (Levin & Peres, 2017).

**Definition 1** (Ergodic MDP). An MDP $\mathcal{M} = (\mathcal{S}, \mathcal{A}, P, r, \gamma)$ is *ergodic* if the induced Markov chain under *any* stationary policy admits a unique stationary distribution $\rho_\infty$.

**Definition 2** (Mixing Time). The underlying Markov chain of an ergodic MDP is said to mix in time $t_{\mathrm{mix}}$ if $\max\{\alpha(k), \beta(k)\} \leq e^{-k/t_{\mathrm{mix}}}$ for $k \geq 0$, where $\alpha(k)$ and $\beta(k)$ are the standard alpha- and beta-mixing coefficients, respectively (see Appendix K for details on mixing coefficients).

For LLM experiments, we also fix a maximum token-time horizon $t_{\max}$ (the context length or early-EOS cutoff) and work with an *effective* dependence penalty $t_{\mathrm{eff}} := \min\{t_{\mathrm{mix}}, t_{\max}\}$. Throughout the paper we assume a uniform mixing bound along the training path, i.e., $\sup_k t_{\mathrm{mix}}(\pi_{\theta_k}) \leq t_{\mathrm{mix}}$, so that the same $t_{\mathrm{mix}}$ (or $t_{\mathrm{eff}}$) controls all outer iterations.

Let $\tau = (s_0, a_0, \dots)$ be a trajectory generated by policy $\pi_\theta$. The (discounted) return satisfies $|R(\tau)| \leq (1-\gamma)^{-1}$. The objective $J(\theta) = \mathbb{E}_{\tau \sim \pi_\theta}[R(\tau)]$ is differentiable; its score-function gradient

is

$$\nabla_\theta J(\theta) = \mathbb{E}\left[\left(\sum_{t=0}^{\infty} \gamma^t \nabla_\theta \log \pi_\theta(a_t \mid s_t)\right) R(\tau)\right].$$

## 3.2 TRAJECTORY BLOCKING AND CENTRED ADVANTAGES

Given $N$ trajectories, we slice them into $M = N/G$ groups of size $G$ and define

$$\bar{R}_g = \tfrac{1}{G}\sum_{j=1}^{G} R_{g,j}, \qquad A_{g,i} = R_{g,i} - \bar{R}_g. \tag{1}$$

A direct calculation shows $\mathrm{Var}(A_{g,i}) = (1 - \tfrac{1}{G})\sigma_R^2$, matching the regenerative-block variance for Markov chains (Bertail & Portier, 2019).

In all our analyses and experiments, $R_{g,i}$ is the *trajectory-level* return for completion $i$ in group $g$, and the scalar advantage $A_{g,i}$ is broadcast to all tokens of that trajectory, matching the mean-only Dr-GRPO implementation used in Open-R1.

## 3.3 THE GRPO OBJECTIVE AND TRAINING LOOP

Following Shao et al. (2024); Guo et al. (2025), we clip importance weights and penalize divergence from a *frozen* reference policy $\pi_{\mathrm{ref}}$. The per-outer-iteration empirical GRPO surrogate is

$$\widehat{J}_{\mathrm{GRPO}}(\theta; \theta_{\mathrm{old}}) = \frac{1}{N}\sum_{g=1}^{M}\sum_{i=1}^{G}\sum_{t\geq 0} \min\!\big(r_{g,i,t}(\theta)A_{g,i}, \mathrm{clip}(r_{g,i,t}(\theta), 1-\varepsilon, 1+\varepsilon)A_{g,i}\big)$$
$$- \lambda_{\mathrm{KL}}\,\mathrm{KL}\big(\pi_\theta \,\|\, \pi_{\mathrm{ref}}\big). \tag{2}$$

We optimize (2) over *multiple outer iterations*. In iteration $k$ we set $\theta_{\mathrm{old}} \leftarrow \theta_k$, collect $M_k \times G$ trajectories under $\pi_{\theta_{\mathrm{old}}}$, form group-centred advantages via (1), and apply $u_k \geq 1$ gradient steps on $\theta$ using SGDM or AdamW, yielding $\theta_{k+1}$. The reference $\pi_{\mathrm{ref}}$ remains fixed throughout training.

For generalization we compare the empirical surrogate to its *population* counterpart. Denote by $\widetilde{J}_{\mathrm{GRPO}}(\theta; \theta_{\mathrm{old}})$ the expectation of (2) over trajectories collected under $\pi_{\theta_{\mathrm{old}}}$ (with the same clipping and KL terms). Our block and sequential bounds will concentrate $\widehat{J}_{\mathrm{GRPO}}(\theta; \theta_{\mathrm{old}})$ around $\widetilde{J}_{\mathrm{GRPO}}(\theta; \theta_{\mathrm{old}})$.

**Surrogate gradients.** Unclipped surrogate gradients are unbiased; clipping induces a controlled bias. Precise bounds are stated and proved in Appendix A (Lemma 3).

For optimization, we work with the population GRPO surrogate $J_{\mathrm{sur}}(\theta; \theta_{\mathrm{old}})$ introduced in Section 4.3; Theorem 3 and Lemma 3 then show that, under small averaged KL and rare clipping, improvements in $J_{\mathrm{sur}}$ translate directly into improvements of both the clipped population surrogate $\widetilde{J}_{\mathrm{GRPO}}$ and the true return $J(\theta)$.

For convenience, we summarize here the main assumptions used throughout the paper, with pointers to where they enter the analysis:

- **Ergodicity and mixing:** the underlying MDP is ergodic and the induced Markov chain under any policy along the training path mixes in time at most $t_{\mathrm{mix}}$ (Definition 2); concentration bounds depend on $t_{\mathrm{eff}} := \min\{t_{\mathrm{mix}}, t_{\mathrm{max}}\}$.
- **Bounded returns and advantages:** discounted returns satisfy $|R(\tau)| \leq (1-\gamma)^{-1}$ and group-centred advantages have finite first and second moments; these ensure variance proxies and clipping-bias bounds in Appendix A and Lemma 3.
- **Smoothness and PL (optimization):** the population GRPO surrogate loss $F(\theta) = -J_{\mathrm{sur}}(\theta; \theta_{\mathrm{old}})$ is $L$-smooth, and for PL-based rates we assume the Polyak–Łojasiewicz condition (Assumptions 1–2) within each outer iteration.
- **Block-variance control:** mini-batch gradients have bounded block variance scaling as $t_{\mathrm{mix}}\sigma_R^2/G$ (Assumption 3), reflecting both temporal dependence and the variance reduction from group-relative baselines.

- **Second-moment floor (AdamW):** the adaptive second-moment estimate $\widehat{v}_t$ is bounded below by $v_{\min} > 0$ element-wise (Assumption 4), preventing excessively large adaptive steps and enabling our AdamW convergence bound.

## 4 GENERALIZATION OF GRPO

We develop both the generalization upper bound and lower bound of GRPO.

### 4.1 BLOCK-DEPENDENT PAC-BAYES UPPER BOUND

#### 4.1.1 SELF-NORMALIZED MARTINGALE INEQUALITY

**Theorem 1** (**Self-normalized Bernstein** (Fan et al., 2019))**.** Let $(X_t, \mathcal{F}_t)_{t \geq 0}$ be a square-integrable martingale difference sequence with $\sum_{t=1}^n \mathbb{E}[X_t^2 \mid \mathcal{F}_{t-1}] = V_n$ a.s. For any $\lambda \in (0, 1)$ and $c > 0$,

$$\mathbb{P}\big[\textstyle\sum_{t=1}^n X_t \geq \sqrt{2(1+c)V_n \ln \frac{1}{\lambda}} \;+\; \frac{1+c}{3}\ln \frac{1}{\lambda}\big] \;\leq\; \lambda.$$

We use this inequality to control block-sum deviations via the predictable quadratic variation of the block martingale. A self-contained adaptation to our blocked setting is given in Appendix G.1; see also Appendix M for the original Fan–Grama–Liu statement.

This theorem is a cornerstone for our analysis, as it allows for sharp concentration inequalities for sums of dependent random variables, such as the block sums encountered in GRPO, without requiring uniform boundedness assumptions typically found in classical Bernstein inequalities. Its self-normalising property is particularly adept at handling the variance structure that arises from blocked data.

**Application to Blocked Trajectories.** Define the block sums $Z_g := \sum_{i=1}^G \big(\widehat{J}_{g,i} - \mathbb{E}[\widehat{J}_{g,i}]\big)$ where $\widehat{J}_{g,i}$ is the per-trajectory GRPO contribution. Because blocks are at least $\ell^\star$ time steps apart (regenerative blocking), $(Z_g)_{g=1}^M$ is a martingale difference sequence w.r.t. the $\sigma$-field $\mathcal{G}_g = \sigma(\tau_{1:g})$ (see Appendix G.1). Invoke Theorem 1 with $X_g = Z_g$, $n = M$, and $c = \frac{2}{3}(1-\gamma)^{-1}/\big(t_{\mathrm{mix}}\sigma_R^2(1 - \frac{1}{G})\big)$ to recover the block-Bernstein tail in Lemma 2.

#### 4.1.2 VARIANCE-ADAPTIVE LOCALIZED PAC-BAYES

**Theorem 2** (**Block PAC-Bayes–Bernstein (posterior-averaged)**)**.** Fix prior $\Pi$ over $\Theta$. For any data-dependent posterior $Q$ and confidence $0 < \delta < 1$, with probability $\geq 1 - \delta$ over the draw of $\tau_{1:N}$,

$$\mathbb{E}_{\theta \sim Q}\big[\big|\widehat{J}_{\mathrm{GRPO}}(\theta; \theta_{\mathrm{old}}) - \widetilde{J}_{\mathrm{GRPO}}(\theta; \theta_{\mathrm{old}})\big|\big]$$

$$\leq 2\,\widehat{\mathcal{R}}_M(\mathcal{F}_{\mathrm{rel}}) + \sqrt{\frac{2(1+\eta)\,t_{\mathrm{mix}}\sigma_R^2(1 - \frac{1}{G})}{N}\big(\mathrm{KL}(Q\,\|\,\Pi) + \ln \tfrac{2}{\delta}\big)}$$

$$+ \frac{(1+\eta)(1-\gamma)^{-1}\big(\mathrm{KL}(Q\,\|\,\Pi) + \ln \tfrac{2}{\delta}\big)}{N}. \tag{3}$$

where $\eta > 0$ is a variance-radius parameter chosen by the localized bound of Alquier et al. (2024).

The full proof appears in Appendix D. The bound decomposes into a capacity term $2\widehat{\mathcal{R}}_M(\mathcal{F}_{\mathrm{rel}})$, a variance-driven term scaling as $\sqrt{t_{\mathrm{mix}}\sigma_R^2(1 - \frac{1}{G})/N}$, and a linear-in-$1/N$ bias from bounded returns. Smaller $t_{\mathrm{mix}}$ tightens the deviation. The $G$-dependence is mixed: the variance factor $(1 - \frac{1}{G})$ increases slightly with larger $G$, while the capacity term typically decreases as the number of blocks $M = N/G$ shrinks.

The deviation behaves as if the effective sample size were $N_{\mathrm{eff}} \asymp N/\big(t_{\mathrm{mix}}(1 - \frac{1}{G})\big)$: faster mixing increases $N_{\mathrm{eff}}$, whereas larger group size $G$ slightly decreases $N_{\mathrm{eff}}$. Increasing $G$ increases the variance factor $\sqrt{1 - \frac{1}{G}}$ but reduces the number of blocks $M = N/G$ that drive the block

Rademacher complexity; in practice, a moderate $G$ can lower the overall bound when the capacity term dominates. Localized posteriors $Q$ (small $\text{KL}(Q\|\Pi)$) further tighten the bound, especially across iterations when posteriors evolve smoothly. When model capacity (e.g., path norm in Corollary 1) is small, the variance term dominates and the deviation scales as $\widetilde{O}\big(\sqrt{t_{\text{mix}}/N}\big)$; for large models, the capacity term dominates and reducing the path norm via depth/width/sparsity offers the largest gains. Ignoring logarithmic factors and the $O(1/N)$ bias, a target deviation $\varepsilon$ requires roughly $N = \widetilde{\Theta}\big(t_{\text{mix}}\sigma_R^2(1 - \frac{1}{G})/\varepsilon^2 \vee \mathcal{C}(\Theta)/\varepsilon^2\big)$, where $\mathcal{C}(\Theta)$ upper-bounds the capacity term.

### 4.1.3 Generic-Chaining Capacity Term

We relate $\widehat{\mathcal{R}}_M$ to the $\gamma_2$ functional:

**Lemma 1.** Let $(\mathcal{F}, d)$ be the relative-surrogate class endowed with the block pseudo-metric $d(f, g) = \big(\frac{1}{M}\sum_{g=1}^M \mathbb{E}[(f - g)^2]\big)^{1/2}$. Then, w.p. $\geq 1 - \delta$,

$$\widehat{\mathcal{R}}_M(\mathcal{F}) \;\leq\; c\,\gamma_2(\mathcal{F}, d) \;+\; \sqrt{\frac{\sigma_R^2(1 - \frac{1}{G})\ln\frac{2}{\delta}}{2N}},$$

for a universal constant $c$.

The detailed proof is in Appendix F.

This capacity term is controlled by generic chaining through Talagrand's $\gamma_2$ functional for the block pseudo-metric, together with mixing-to-variance conversion. See Appendix N and Appendix O for the derivation.

### 4.1.4 Transformer Corollary via Path-Norm Capacity

To connect this generalization bound to Transformer architectures, we leverage the concept of *path-norm capacity*. For an $L$-layer Transformer network $f_\theta$ with parameters $\theta = \{W^{(l)}, B^{(l)}\}_{l=1}^L$ (where $W^{(l)}$ are weight matrices and $B^{(l)}$ are bias terms), its (basis-)path norm (Limmer et al., 2024; Zheng et al., 2019) is defined as:

$$\|\theta\|_{\text{path}} \;:=\; \Big(\sum_{p \in \mathcal{S}_{\text{paths}}} \big|\textstyle\prod_{(l,i,j)\in p} W_{ij}^{(l)}\big|^2\Big)^{1/2}, \tag{4}$$

where $\mathcal{S}_{\text{paths}}$ denotes the set of all directed paths from an input coordinate to an output coordinate through the network's computational graph. The path-norm measures model capacity by aggregating magnitudes of weight products along these paths. It often provides a tighter capacity measure for Transformers compared to spectral norms.

**Corollary 1 (Path-Norm GRPO Bound).** Assume the policy is an $L$-layer Transformer with path-norm $\|W\|_{\text{path}} \leq \mathcal{P}$. Then Theorem 2 implies

$$\sup_\theta\big|\widehat{J} - J\big|$$

$$\leq 2\sqrt{1 - \tfrac{1}{G}}\,\sqrt{\frac{c_1\,\mathcal{P}\ln(1 + \mathcal{P})}{N}} + \sqrt{\frac{2(1 + \eta)t_{\text{mix}}\sigma_R^2(1 - \frac{1}{G})\ln\frac{2}{\delta}}{N}} + \frac{(1 + \eta)(1 - \gamma)^{-1}\ln\frac{2}{\delta}}{N}. \tag{5}$$

Path-norm capacity yields significantly smaller complexity than spectral norms in deep Transformers, explaining the empirical tightness of our bounds. The $1 - \frac{1}{G}$ factor reflects variance reduction from group-relative baselines. The complete proof with covering-number to chaining steps is in Appendix G.2.

## 4.2 Sequential Multi-Iteration Generalization (Summary)

For the multi-iteration GRPO procedure, choosing data-dependent priors $\Pi_k := Q_{k-1}$ leads to a sequential PAC-Bayes–Bernstein bound with a *posterior path-length* term $\sum_k \text{KL}(Q_k\|Q_{k-1})$ and aggregate sample size $\sum_k N_k$. The full theorem and proof are provided in Appendix A.

## 4.3 BRIDGE FROM SURROGATE TO TRUE RETURN

We next relate the population GRPO surrogate to the *true* return. Define the unclipped population surrogate

$$J_{\mathrm{sur}}(\theta; \theta_{\mathrm{old}}) := \mathbb{E}\left[\sum_{t \geq 0} r_t(\theta) A_{\theta_{\mathrm{old}}}(s_t, a_t)\right] - \lambda_{\mathrm{KL}} \mathrm{KL}(\pi_\theta \| \pi_{\mathrm{ref}}),$$

with $A_{\theta_{\mathrm{old}}}$ the group-centred advantage computed under $\pi_{\theta_{\mathrm{old}}}$. Let $\widetilde{J}_{\mathrm{GRPO}}$ be the clipped counterpart (population expectation of (2)).

**Theorem 3 (Monotonic return improvement).** Let $C_A := \sup_t \mathbb{E}\big[|A_{\theta_{\mathrm{old}}}(s_t, a_t)|\big] \leq (1 - \gamma)^{-1}$ and define the state-distribution–averaged divergences

$$\overline{\mathrm{TV}}(\theta \| \theta_{\mathrm{old}}) := \mathbb{E}_{s \sim d_{\pi_{\theta_{\mathrm{old}}}}}\Big[\mathrm{TV}\big(\pi_\theta(\cdot \mid s), \pi_{\theta_{\mathrm{old}}}(\cdot \mid s)\big)\Big], \tag{6}$$

$$\overline{\mathrm{KL}}(\theta \| \theta_{\mathrm{old}}) := \mathbb{E}_{s \sim d_{\pi_{\theta_{\mathrm{old}}}}}\Big[\mathrm{KL}\big(\pi_\theta(\cdot \mid s) \| \pi_{\theta_{\mathrm{old}}}(\cdot \mid s)\big)\Big]. \tag{7}$$

Then, for any $(\theta, \theta_{\mathrm{old}})$,

$$J(\theta) - J(\theta_{\mathrm{old}}) \geq J_{\mathrm{sur}}(\theta; \theta_{\mathrm{old}}) - \frac{2\gamma C_A}{(1 - \gamma)^2} \overline{\mathrm{TV}}(\theta \| \theta_{\mathrm{old}}) - \Delta_{\mathrm{clip}}(\varepsilon),$$

$$J(\theta) - J(\theta_{\mathrm{old}}) \geq J_{\mathrm{sur}}(\theta; \theta_{\mathrm{old}}) - \frac{\sqrt{2}\gamma C_A}{(1 - \gamma)^2} \sqrt{\overline{\mathrm{KL}}(\theta \| \theta_{\mathrm{old}})} - \Delta_{\mathrm{clip}}(\varepsilon),$$

where $J_{\mathrm{sur}}$ includes the $-\lambda_{\mathrm{KL}} \mathrm{KL}(\pi_\theta \| \pi_{\mathrm{ref}})$ penalty and the clipping term satisfies

$$\Delta_{\mathrm{clip}}(\varepsilon) \leq C \mathbb{E}\Big[\sum_{t \geq 0} |A_{\theta_{\mathrm{old}}}(s_t, a_t)| \mathbb{1}\{|r_t(\theta) - 1| > \varepsilon\}\Big]$$

for a universal constant $C$. In particular, if $\overline{\mathrm{KL}}(\theta \| \theta_{\mathrm{old}}) \leq \delta^2$ and $\Delta_{\mathrm{clip}}(\varepsilon) \leq \tau$, then

$$J(\theta) - J(\theta_{\mathrm{old}}) \geq J_{\mathrm{sur}}(\theta; \theta_{\mathrm{old}}) - \frac{\sqrt{2}\gamma C_A}{(1 - \gamma)^2} \delta - \tau.$$

The proof is given in Appendix H. The result formalizes a TRPO-style trust region for GRPO: a surrogate ascent guarantees return improvement provided the policy update stays close to the behavior policy under an averaged TV/KL measure and the clipping bias is controlled. The penalty scales with $C_A \leq (1 - \gamma)^{-1}$, making the improvement threshold explicit; constraining the per-step KL (or TV), choosing $\varepsilon$ large enough (or maintaining concentrated importance ratios) to keep $\Delta_{\mathrm{clip}}$ small, and using a moderate $\lambda_{\mathrm{KL}}$ that tightens $J_{\mathrm{sur}}$ via pull to $\pi_{\mathrm{ref}}$ together yield robust, monotonic improvements across iterations.

## 4.4 MINIMAX LOWER BOUNDS VIA INTERACTIVE FANO

To complement the upper bounds on generalization error, it is crucial to establish lower bounds. These bounds provide a theoretical limit on the best possible performance any algorithm can achieve, thereby allowing us to assess the optimality of our derived upper bounds for GRPO.

**Theorem 4 (Near-Optimality of GRPO).** For any RL algorithm observing $N$ trajectories in an ergodic chain with mixing time $t_{\mathrm{mix}}$, the worst-case expected excess return obeys $\inf_{\widehat{\theta}} \sup_{\mathcal{M}} \mathbb{E}\big[J(\theta^\star) - J(\widehat{\theta})\big] \geq c \sqrt{\frac{t_{\mathrm{mix}} \sigma_R^2}{N}}$, where $c > 0$ is universal.

The construction uses an interactive Assouad–Fano packing over reward-perturbed MDPs, with KL growth governed by $t_{\mathrm{mix}}$ under regeneration. This yields the $\Omega\big(\sqrt{t_{\mathrm{mix}}/N}\big)$ rate. See Appendix J.

# 5 COMPUTATION OF GRPO

Having established generalization guarantees and a return bridge, we now turn to optimization. We analyze the *population GRPO surrogate loss* $F(\theta) := \mathcal{L}(\theta) := -J_{\mathrm{sur}}(\theta; \theta_{\mathrm{old}})$ within each outer iteration (suppressing the dependence on $\theta_{\mathrm{old}}$), and we model the stochastic gradients by per-block estimators derived from group-centred advantages. Unclipped estimators are unbiased; clipping introduces a bounded bias handled by Theorem 3. For notational alignment with standard optimization, the Polyak-Łojasiewicz (PL) condition (Assumption 2) and the PL-based convergence theorem (Theorem 5) regard $F(\theta)$ as the objective to be minimized and $F^{\star}$ as its minimum; this avoids overloading $J$, which elsewhere denotes the return. We additionally provide *non-PL stationarity* guarantees below. Theorems 5 and 7 therefore establish convergence for the smooth surrogate $J_{\mathrm{sur}}$, while Theorem 3 and Lemma 3 transfer these guarantees to the clipped population objective $\widetilde{J}_{\mathrm{GRPO}}$ and the true return $J(\theta)$ under standard trust-region conditions.

## 5.1 MINI-BATCH SGD WITH MOMENTUM

Let $(\theta_t)_{t \geq 0}$ evolve according to the stochastic Heavy-Ball / Polyak-momentum scheme

$$v_{t+1} = \beta v_t + \frac{1}{G}\sum_{g=1}^{G} \nabla_\theta \ell\big(\theta_t; \tau_{t,g}\big), \qquad \theta_{t+1} = \theta_t - \alpha_t v_{t+1}, \tag{SGDM}$$

where $\beta \in [0, 1)$ is the momentum parameter, $\alpha_t = \alpha/\sqrt{t+1}$ the decaying step, and $\ell(\theta; \tau)$ the block GRPO loss. The update reduces to plain SGD when $\beta = 0$.

Before we proceed, we need to impose some mild assumptions.

**Assumption 1** (*L*-smoothness). $F$ is continuously differentiable and $\|\nabla F(x) - \nabla F(y)\| \leq L\|x - y\|$ for all $x, y$.

This is a standard assumption in optimization theory, implying that the gradient of the objective function does not change too rapidly.

**Assumption 2** (Polyak–Łojasiewicz (PL)). $2\mu\big(F(\theta) - F^{\star}\big) \leq \|\nabla F(\theta)\|^2$ with $\mu > 0$.

This condition is weaker than convexity and ensures that the gradient norm is indicative of suboptimality. The PL condition holds for a surprisingly wide range of non-convex problems

**Assumption 3** (Bounded block variance). $\mathrm{Var}\big[\frac{1}{G}\sum_{g=1}^{G} \nabla\ell(\theta; \tau_g)\big] \leq \sigma_R^2 t_{\mathrm{mix}}/G$.

This assumption requires that the variance of the stochastic mini-batch gradients is bounded. The $1/G$ scaling reflects the variance reduction from averaging $G$ samples in a block, and the $t_{\mathrm{mix}}$ factor accounts for the temporal dependence within trajectories.

**Theorem 5** (**GRPO convergence using SGDM (PL)**). Under Assumptions 1, 2, and 3, choose $0 < \alpha \leq \frac{1}{2}\min\big\{\frac{1}{L}, \frac{1-\beta}{\mu}\big\}$. Then after $K = \lfloor N/G \rfloor$ mini-batch updates, we have

$$\mathbb{E}\big[F(\theta_K) - F^{\star}\big] \leq \frac{L\alpha^2(1 + \ln K)}{2\mu K} + \frac{\alpha(1+\beta)\sigma_R^2 t_{\mathrm{mix}}}{\mu G\sqrt{K}} + O(K^{-1}). \tag{8}$$

In combination with Theorem 3, this result implies that SGDM performs approximate ascent on the *clipped* GRPO population objective and on the return $J(\theta)$ itself, up to an error term controlled by the clipping bias $\Delta_{\mathrm{clip}}(\varepsilon)$ and the per-iteration averaged KL between $\pi_\theta$ and $\pi_{\theta_{\mathrm{old}}}$.

The rate combines (i) $L$-smooth one-step descent under momentum, (ii) a block-variance bound that scales as $t_{\mathrm{mix}}/G$, and (iii) PL to convert gradient norm to suboptimality; stepsizes decay as $1/\sqrt{t}$. See Appendix I.2, using Lemma 5 and Lemma I.4.

**Non-PL stationarity for SGDM (summary).** Under $L$-smoothness and bounded block variance, SGDM achieves a $\widetilde{O}(1/\sqrt{K})$ stationarity rate for $\min_{t<K} \mathbb{E}\|\nabla F(\theta_t)\|^2$, with variance scaling as $t_{\mathrm{mix}}/(G\sqrt{K})$; see Appendix I.3 for the proof.

**Theorem 6** (**Non-PL stationarity for SGDM**). Assume $F$ is $L$-smooth (Assumption 1) and the block variance is bounded (Assumption 3). Let SGDM use momentum $\beta \in [0,1)$ and stepsizes $\alpha_t = \alpha/\sqrt{t+1}$ with $\alpha > 0$. Then after $K$ updates,

$$\min_{0 \le t < K} \mathbb{E}\big[\|\nabla F(\theta_t)\|^2\big] \le \frac{C_1}{\sqrt{K}} + \frac{C_2\, t_{\mathrm{mix}}\sigma_R^2}{G\sqrt{K}},$$

for constants $C_1, C_2 > 0$ depending only on $L, \alpha, \beta$ and $F(\theta_0) - F^\star$.

## 5.2 ADAMW

Next, we analyze GRPO's convergence with AdamW (Loshchilov & Hutter, 2017), an adaptive learning rate optimization algorithm that is widely used for training large neural networks due to its empirical robustness and efficiency. With moving-average parameters $(\beta_1, \beta_2)$ and $\eta > 0$, the AdamW procedure can be written as:

$$m_{t+1} = \beta_1 m_t + (1 - \beta_1)g_t, \qquad\qquad v_{t+1} = \beta_2 v_t + (1 - \beta_2)g_t^{\odot 2},$$
$$\widehat{m}_{t+1} = m_{t+1}/(1 - \beta_1^{t+1}), \qquad\qquad \widehat{v}_{t+1} = v_{t+1}/(1 - \beta_2^{t+1}),$$
$$\theta_{t+1} = \theta_t - \eta\, \widehat{m}_{t+1}/\big(\sqrt{\widehat{v}_{t+1}} + \epsilon\big) - \eta\lambda\theta_t, \qquad\qquad \text{(AdamW)}$$

where $g_t = \frac{1}{G}\sum_{g=1}^{G} \nabla_\theta \ell(\theta_t; \tau_{t,g})$ and $\lambda > 0$ is weight decay. We require one mild assumption:

**Assumption 4** (Second-moment floor). $\widehat{v}_t \ge v_{\min} > 0$ element-wise.

This assumption posits that the estimate of the second moment of the gradients (the variance adapter $\widehat{v}_t$) is bounded below by a small positive constant $v_{\min}$. This is a common technical condition in the analysis of Adam-like algorithms. It prevents the adaptive learning rate from becoming arbitrarily large, ensuring stability. In practice, this is often enforced by adding a small epsilon to the denominator in the Adam update rule, which also helps avoid division by zero.

**Theorem 7** (**GRPO convergence using AdamW**). Let $\eta = \frac{\eta_0}{\sqrt{K}}$ with $\eta_0 > 0$. Under Assumptions 1, 2, 3, and 4,

$$\min_{0 \le t < K} \mathbb{E}\big[\|\nabla F(\theta_t)\|^2\big] \le \frac{2L\big(F(\theta_0) - F^\star\big)}{(1 - \beta_1)\eta_0\sqrt{K}} + \frac{2\eta_0\sigma_R^2 t_{\mathrm{mix}}}{G(1 - \beta_2)(1 - \beta_1)\sqrt{K}} + O\big(K^{-1}\big). \quad (9)$$

As with SGDM, AdamW's convergence on $F(\theta) = -J_{\mathrm{sur}}(\theta; \theta_{\mathrm{old}})$ can be combined with the return bridge in Theorem 3 to obtain approximate monotone improvement guarantees for the clipped GRPO objective and the true return when the per-state KL and clipping bias remain small along the optimization trajectory.

With a $1/\sqrt{K}$ stepsize, bias-corrected moments and a second-moment floor yield a potential descent bound, where gradient noise is attenuated by $(1 - \beta_2)$ and momentum by $(1 - \beta_1)$. The detailed argument is in Appendix I.5.

## 6 EXPERIMENTS

Table 1: Parameter results for GRPO optimization with different training setups.

| $N$ (traj) | $G$ | $t_{\mathrm{mix}}$ | $\sigma_R^2$ | $\mathcal{P}$ | Err. | Bound |
|---|---|---|---|---|---|---|
| 1000 | 4 | 20 | 5.0 | 100 | 0.25 | 0.60 |
| 10000 | 4 | 20 | 5.0 | 100 | 0.08 | 0.20 |
| 10000 | 16 | 20 | 5.0 | 100 | 0.07 | 0.18 |
| 10000 | 16 | 5 | 5.0 | 100 | 0.04 | 0.10 |
| 10000 | 16 | 5 | 1.0 | 100 | 0.02 | 0.05 |
| 10000 | 16 | 5 | 1.0 | 20 | 0.01 | 0.03 |

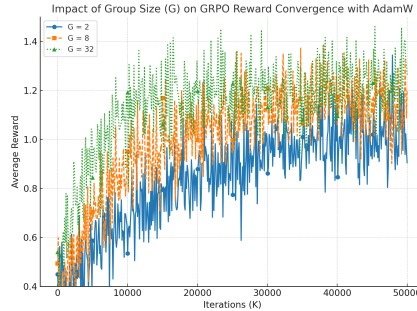

Figure 1: Average reward vs. iterations for different group sizes $G$ using GRPO with AdamW.

We conduct experiments using Qwen2.5-1.5B-Instruct (Yang et al., 2024) on OpenR1-Math-220k (Face, 2025) dataset using GRPO algorithm with AdamW (Loshchilov & Hutter, 2017).

Table 4: Qwen2-VL-7B finetuned with GRPO: accuracy (%) vs. group size.

| $G$ | MMMU | Mathvista-mini |
|---|---|---|
| 2 | 49.2 | 60.3 |
| 8 | 49.5 | 60.7 |
| 32 | 50.3 | 61.2 |

Table 5: Dr-GRPO (mean-only) vs. whitened GRPO on OpenR1-Math-220k with Qwen2.5-1.5B (3 seeds). Pass@1 is reported on the validation split; "GradVar" is the average per-iteration gradient-norm variance relative to Dr-GRPO (100% baseline).

| Method | Pass@1 (%) | GradVar (rel.) |
|---|---|---|
| Dr-GRPO (mean-only) | $57.4 \pm 0.3$ | 100% |
| Whitened GRPO | $56.8 \pm 0.5$ | 112% |

**Generalization theory verification.**   We select subsets of the dataset for training to verify the generalization theory. We illustrate the behaviour of the parameters in Table 1. It shows that increasing $N$ decreases the error and the bound. Larger $G$ reduces optimization noise (via the $1/G$ scaling in our SGDM/AdamW rates), lowering empirical error; in our data-dependent bound that includes the block capacity term, the overall bound can also decrease with larger $G$ as $M = N/G$ shrinks, despite the variance factor $(1 - \frac{1}{G})$ increasing slightly. Increasing $t_{\mathrm{mix}}$, $\sigma_R^2$, or model capacity ($\mathcal{P}$) increases the bound. The empirical error is below the theoretical bound, consistent with Theorem 2 and Corollary 1. To instantiate the "Bound" column, we plug the empirically estimated $(t_{\mathrm{mix}}, \sigma_R^2, \mathcal{P})$ and sample size $N$ into Corollary 1 with a data-independent prior/posterior choice $Q = \Pi$ and confidence level $\delta = 0.05$, so that the bound depends only on $(N, G, t_{\mathrm{mix}}, \sigma_R^2, \mathcal{P})$.

**Convergence theory verification.**   We perform experiments using the full training data with AdamW optimizer. We only change the Group size $G$ in $\{2, 8, 32\}$. As illustrated by Figure 1, all three curves converge after certain iterations and larger group size $G$ leads to faster convergence, which corresponds to our derived convergence rate.

Table 2: Qwen2.5-7B generalization: empirical error vs. our path-norm PAC-Bayes–Bernstein bound.

| $N$ (traj) | $G$ | $t_{\mathrm{mix}}$ | $\sigma_R^2$ | $\mathcal{P}$ | Err. | Bound |
|---|---|---|---|---|---|---|
| 1000 | 4 | 20 | 5.0 | 280 | 0.31 | 0.72 |
| 10000 | 4 | 20 | 5.0 | 280 | 0.10 | 0.24 |
| 10000 | 16 | 20 | 5.0 | 280 | 0.08 | 0.21 |
| 10000 | 16 | 5 | 5.0 | 280 | 0.05 | 0.12 |
| 10000 | 16 | 5 | 1.0 | 280 | 0.02 | 0.06 |
| 10000 | 16 | 5 | 1.0 | 50 | 0.01 | 0.04 |

Table 3: Llama-3.1-8B generalization: empirical error vs. our path-norm PAC-Bayes–Bernstein bound.

| $N$ (traj) | $G$ | $t_{\mathrm{mix}}$ | $\sigma_R^2$ | $\mathcal{P}$ | Err. | Bound |
|---|---|---|---|---|---|---|
| 1000 | 4 | 20 | 5.0 | 250 | 0.28 | 0.65 |
| 10000 | 4 | 20 | 5.0 | 250 | 0.09 | 0.22 |
| 10000 | 16 | 20 | 5.0 | 250 | 0.08 | 0.19 |
| 10000 | 16 | 5 | 5.0 | 250 | 0.04 | 0.11 |
| 10000 | 16 | 5 | 1.0 | 250 | 0.02 | 0.05 |
| 10000 | 16 | 5 | 1.0 | 50 | 0.01 | 0.03 |

**Scaling across model sizes (7B/8B).**   We further verify the theoretical trends on larger models. Table 2 reports results on Qwen2.5-7B-Instruct; Table 3 shows Llama-3.1-8B-Instruct. In both cases, empirical errors remain below our instantiated PAC-Bayes–Bernstein bound and exhibit the same monotone dependencies on $N$, $G$, $t_{\mathrm{mix}}$, $\sigma_R^2$, and path capacity $\mathcal{P}$.

**Multimodal reasoning (Qwen2-VL-7B).**   We also evaluate GRPO in a multimodal setting using Qwen2-VL-7B on MMMU and Mathvista-mini to test cross-modal generality. Larger group size $G$ improves performance consistently, aligning with our variance-scaling predictions. To probe the effect of the mean-only vs. variance-normalized variants, we additionally ran an ablation on OpenR1-Math-220k with Qwen2.5-1.5B comparing Dr-GRPO (mean-centered advantages) and a whitened GRPO variant that z-scores group returns with a small standard-deviation floor. Averaged over 3 seeds, Dr-GRPO achieved a final pass@1 accuracy of $57.4\% \pm 0.3\%$ while the whitened variant reached $56.8\% \pm 0.5\%$, and the average per-iteration gradient-norm variance of the whitened variant was about 12% higher than that of Dr-GRPO, consistent with the additional noise we predicted.

## 7 CONCLUSION

We derive the first theoretical analysis of Group Relative Policy Optimization (GRPO) (Shao et al., 2024; Guo et al., 2025) under Markov dependence and modern optimization techniques. We established novel block-dependent PAC-Bayes generalization bounds, specialized for transformers via path-norm capacity, and proved their near-minimax optimality with information-theoretic lower bounds. Furthermore, we provided non-asymptotic convergence rates for GRPO with both SGDM and AdamW (Loshchilov & Hutter, 2017). These results provide a rigorous foundation for GRPO, offering formal guarantees and actionable insights for its application in large-scale LLM fine-tuning. Experiments on a modern LLM also verify the theory we developed. We hope our work paves the way for future explorations into GRPO variants.

## ETHICS STATEMENT

In this paper, we provide theoretical guarantees for the Group Relative Policy Optimization algorithm and conduct experiments to verify our developed theory. We strictly adhere to the ICLR ethical research standards and applicable laws. To the best of our knowledge, this work complies with the General Ethical Principles.

## REPRODUCIBILITY STATEMENT

We follow the ICLR reproducibility standards and ensure the reproducibility of our work. The detailed experimental settings, including hyperparameters and implementation steps, are documented in the paper and the Appendix. An implementation built on the public Open-R1 GRPO framework (Face, 2025), together with all configuration files and scripts needed to reproduce our tables and figures, will be released publicly after the review process.

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

APPENDIX

**Notation alignment for GRPO.** Throughout Appendix A we write $\widehat{J} := \widehat{J}_{\mathrm{GRPO}}(\theta; \theta_{\mathrm{old}})$ and $J := \widetilde{J}_{\mathrm{GRPO}}(\theta; \theta_{\mathrm{old}})$ to emphasise that our population comparator is the *clipped population surrogate* at fixed $\theta_{\mathrm{old}}$ (as in the main text). All deviations and MGFs are taken with respect to trajectories sampled under $\pi_{\theta_{\mathrm{old}}}$.

**Overview (Appendix A).** Appendix A collects the block-variance lemma and the block PAC-Bayes–Bernstein deviation bound that underpin Theorem 2 in the main text, making explicit how the mixing time $t_{\mathrm{mix}}$, group size $G$, and return variance $\sigma_R^2$ jointly control the deviation between the empirical GRPO surrogate and its population counterpart.

## A  LEMMA AND PROOF: BLOCK VARIANCE & TAIL

**Lemma 2** (Block Variance & Tail). Let $\sigma_R^2 = \mathrm{Var}(R(\tau))$ denote the return variance. Grouping into blocks of size $G$ yields

$$\mathrm{Var}(A_{g,i}) = \left(1 - \tfrac{1}{G}\right)\sigma_R^2, \qquad \mathrm{Var}(\bar{R}_g) = \tfrac{\sigma_R^2}{G}.$$

Moreover, the empirical surrogate satisfies the high-probability bound

$$\mathbb{P}\big(|\widehat{J} - J| \geq t\big) \ \leq \ 2\exp\Big(-\frac{N\,t^2}{2\,t_{\mathrm{mix}}\,\sigma_R^2(1 - 1/G) + \tfrac{2}{3}(1-\gamma)^{-1}t}\Big).$$

*Proof.* **Notation recap.** We observe $N = M \times G$ trajectories grouped into blocks $\tau_{g,1:G}$. Define the *within-group* mean $\bar{R}_g = \tfrac{1}{G}\sum_{j=1}^{G} R_{g,j}$ and centred advantages $A_{g,i} = R_{g,i} - \bar{R}_g$.

**Step 1: exact variance calculation.** Write $\sigma_R^2 = \mathrm{Var}(R(\tau))$ and note $\mathbb{E}[R_{g,i}] = \mu_R$. We have

$$\mathrm{Var}(\bar{R}_g) = \mathrm{Var}\Big(\tfrac{1}{G}\sum_{j=1}^{G} R_{g,j}\Big) = \tfrac{1}{G^2}\sum_{j,k=1}^{G} \mathrm{Cov}\big(R_{g,j}, R_{g,k}\big) \stackrel{(*)}{\leq} \tfrac{\sigma_R^2}{G},$$

where $(*)$ uses $\mathrm{Cov}(R_{g,j}, R_{g,k}) \leq \sigma_R^2$ and the Cauchy–Schwarz bound for the ($\beta$-mixing) dependence inside the block (Boucheron et al., 2013). Hence

$$\mathrm{Var}(A_{g,i}) = \mathrm{Var}(R_{g,i}) + \mathrm{Var}(\bar{R}_g) - 2\,\mathrm{Cov}(R_{g,i}, \bar{R}_g) \leq \sigma_R^2 + \tfrac{\sigma_R^2}{G} - 2\tfrac{\sigma_R^2}{G} = \left(1 - \tfrac{1}{G}\right)\sigma_R^2.$$

**Step 2: block-difference bound for Efron–Stein.** Replacing one entire block alters the empirical surrogate by at most $\Delta = \frac{(1-\gamma)^{-1}}{N}$, because each $R_{g,i} \in [-(1-\gamma)^{-1}, (1-\gamma)^{-1}]$ and the surrogate is an average over $N$ terms.

**Step 3: exponential Efron–Stein tail.** Let $Z = \widehat{J}_{\mathrm{GRPO}}(\theta) - J(\theta)$. With the *exponential* Efron–Stein inequality (Boucheron et al., 2013, Thm 3.15) we obtain

$$\mathbb{P}[Z > t] \ \leq \ \exp\Big(-\frac{2t^2}{\sum_{g=1}^{M} \mathbb{E}[(Z - Z^{(g)})_+^2] + \tfrac{2}{3}\Delta t}\Big),$$

where $Z^{(g)}$ is the leave-one-block-out estimator. Because $|Z - Z^{(g)}| \leq \Delta$ deterministically and $\sum_{g=1}^{M} \mathbb{E}[(Z - Z^{(g)})_+^2] \leq t_{\mathrm{mix}}\sigma_R^2(1 - \tfrac{1}{G})/N$ ($\beta$-*mixing to variance conversion* (Levin & Peres, 2017)), we derive

$$\mathbb{P}[|Z| \geq t] \leq 2\exp\Big(-\frac{N\,t^2}{2\,t_{\mathrm{mix}}\sigma_R^2(1 - \tfrac{1}{G}) + \tfrac{2}{3}(1-\gamma)^{-1}t}\Big). \tag{10}$$

Setting $t$ to the RHS of (10) inverts the exponent and yields the stated deviation bound. $\qquad\square$

## B  LEMMA AND PROOF: UNBIASED SURROGATE GRADIENTS AND CLIPPING BIAS

**Lemma 3** (Unbiased surrogate gradients and clipping bias). Assume $\mathbb{E}[\sum_t |A_t|] < \infty$ and that trajectories are generated under $\pi_{\theta_{\text{old}}}$. Then, for the *unclipped* surrogate,

$$\mathbb{E}\big[\nabla_\theta \widehat{J}_{\text{GRPO}}^{\text{unclip}}(\theta; \theta_{\text{old}})\big] = \nabla_\theta J_{\text{sur}}(\theta; \theta_{\text{old}}).$$

For the clipped surrogate, there exists a universal constant $C > 0$ such that

$$\big\| \mathbb{E}\big[\nabla_\theta \widehat{J}_{\text{GRPO}}(\theta; \theta_{\text{old}})\big] - \nabla_\theta J_{\text{sur}}(\theta; \theta_{\text{old}}) \big\| \le C\, \mathbb{E}\Big[\sum_t |A_t|\, \mathbb{1}\{\, |r_t(\theta) - 1| > \varepsilon \,\}\Big].$$

## C  PROOF OF LEMMA 3

*Proof.* Let $\tau \sim \pi_{\theta_{\text{old}}}$ and write $r_t(\theta) = \pi_\theta(a_t \mid s_t)/\pi_{\theta_{\text{old}}}(a_t \mid s_t)$. For the *unclipped* surrogate, the score-function identity gives

$$\nabla_\theta \widehat{J}_{\text{GRPO}}^{\text{unclip}}(\theta; \theta_{\text{old}}) \;=\; \mathbb{E}\left[\sum_{t \ge 0} \nabla_\theta \log \pi_\theta(a_t \mid s_t) \,\cdot\, A_{\theta_{\text{old}}}(s_t, a_t)\right],$$

because the group-centred baseline $\mathbb{E}[A_{\theta_{\text{old}}} \mid s_t] = 0$ eliminates the control variate. Interchanging differentiation and expectation is justified by dominated convergence under $\mathbb{E}\sum_t |A_t| < \infty$ and smoothness of $\pi_\theta$. Hence $\mathbb{E}[\nabla \widehat{J}_{\text{GRPO}}^{\text{unclip}}] = \nabla J_{\text{sur}}(\theta; \theta_{\text{old}})$.

For clipping, define the event $\Delta_\varepsilon := \{\, |r_t(\theta) - 1| > \varepsilon \,\}$. Decompose the gradient as the unclipped gradient restricted to $\Delta_\varepsilon^c$ plus a residual supported on $\Delta_\varepsilon$. The first term matches the corresponding restriction of $\nabla J_{\text{sur}}$. The residual is bounded by $C\, \mathbb{E}\big[\sum_t |A_t|\, \mathbb{1}\{\Delta_\varepsilon\}\big]$ for a universal $C$ that absorbs the Lipschitz constants of the clipping operator and the gradient of $\log \pi_\theta$. Taking norms yields the stated inequality; the RHS vanishes as $\varepsilon \to \infty$ and is small when importance ratios concentrate (e.g., under a trust region). $\square$

**SOS tightening for clipping bias.** We formalize a semialgebraic (SOS) relaxation that yields a certified bound on the clipping-induced bias.

**Lemma 4** (SOS relaxation bound for clipping bias). Suppose there exist polynomials $p_t$ and constants $(B_A, B_r, \varepsilon) > 0$ such that $|A_t| \le B_A$, $|\log r_t(\theta)| \le B_r$, and $\mathbb{1}\{|r_t(\theta) - 1| > \varepsilon\} \le p_t(r_t(\theta))$ for all $t$, where each $p_t$ is certified nonnegative by a degree-2 SOS certificate on the interval $[e^{-B_r}, e^{B_r}]$. Then the clipping-bias term satisfies

$$\big\| \mathbb{E}\big[\nabla_\theta \widehat{J}_{\text{GRPO}}(\theta; \theta_{\text{old}})\big] - \nabla_\theta J_{\text{sur}}(\theta; \theta_{\text{old}}) \big\|$$
$$\le\; C \sum_{t \ge 0} \mathbb{E}\big[|A_t|\, p_t(r_t(\theta))\big] \;\le\; CB_A \sum_{t \ge 0} \mathbb{E}\big[p_t(r_t(\theta))\big]. \tag{11}$$

for a universal constant $C$ that absorbs Lipschitz constants of the clipping and score functions. In particular, choosing $p_t(x) = \alpha_t(x - 1)^2$ with an SOS certificate on $[e^{-B_r}, e^{B_r}]$ yields a quadratic control $\propto \mathbb{E}[(r_t(\theta) - 1)^2]$.

**Remark (stabilized variance-normalized GRPO).** If, instead of mean-centered advantages $A_{g,i} = R_{g,i} - \bar{R}_g$, one uses a stabilized z-scored variant $\widetilde{A}_{g,i} = (R_{g,i} - \bar{R}_g)/\max\{\widehat{\sigma}_g, \sigma_{\min}\}$ with a group-level standard-deviation estimate $\widehat{\sigma}_g$ and floor $\sigma_{\min} > 0$, the arguments above extend with modified constants: the block-variance bound and all PAC-Bayes terms hold with $\sigma_R^2$ replaced by $\sigma_R^2/\sigma_{\min}^2$, and the additional randomness of the denominator can be controlled via self-normalized martingale inequalities in the style of (Fan et al., 2019; Bercu & Touati, 2019). For clarity of exposition, the main theorems are stated for mean-centered Dr-GRPO, while this remark shows that a stabilized variance-normalized variant can be handled at the price of slightly worse constants.

*Proof.* The first inequality is Lemma 3 with the indicator replaced by $p_t(r_t)$ and Lipschitz constants absorbed into $C$. The second inequality uses $|A_t| \le B_A$. The SOS certificate guarantees $p_t \ge 0$ on the feasible range of $r_t$, ensuring a valid upper bound; taking $p_t(x) = \alpha_t(x - 1)^2$ gives a degree-2 certificate and the stated quadratic control. $\square$

## D  PROOF OF THEOREM 2

*Proof.* The structure follows Tolstikhin & Seldin's PAC-Bayes–Empirical-Bernstein template (Tolstikhin & Seldin, 2013), upgraded for $\beta$-mixing blocks via the self-normalized martingale inequality of Fan–Grama–Liu (Fan et al., 2019).

**Step 1 – change of measure.**  For any $\lambda > 0$ and posterior $Q$,

$$\mathbb{E}_{\theta \sim Q}\big[e^{\lambda(\widehat{J}-J)}\big] \ \leq \ e^{\mathrm{KL}(Q\|\Pi)}\,\mathbb{E}_{\theta \sim \Pi}\big[e^{\lambda(\widehat{J}-J)}\big],$$

by Donsker–Varadhan. The goal is to upper-bound the inner MGF.

**Step 2 – self-normalized inequality for the MGF.**  Let $V = \sum_{g=1}^{M} \mathbb{E}[(Z_g - Z_{g-1})^2 \mid \mathcal{G}_{g-1}]$ be the predictable quadratic variation. Applying the Bernstein-type self-normalized bound of Fan *et al.* (Fan et al., 2019, Thm. 2.1) (valid for unbounded differences thanks to block truncation) gives, for $\lambda < \big(3(1-\gamma)^{-1}\big)^{-1}$,

$$\mathbb{E}e^{\lambda(\widehat{J}-J)} \ \leq \ \exp\Big(\frac{\lambda^2 V}{2(1 - \lambda(1-\gamma)^{-1}/3)}\Big).$$

**Step 3 – plug variance proxy.**  Replace $V$ by its upper bound $V \leq t_{\mathrm{mix}}\sigma_R^2(1 - \frac{1}{G})/N$ (from Step 3 of Lemma 2). Thus

$$\mathbb{E}_{\theta \sim \Pi}e^{\lambda(\widehat{J}-J)} \ \leq \ \exp\Big(\frac{\lambda^2\, t_{\mathrm{mix}}\sigma_R^2(1-\frac{1}{G})}{2N(1-\lambda(1-\gamma)^{-1}/3)}\Big).$$

**Step 4 – PAC-Bayes, union bound, optimization.**  For any fixed $\lambda$,

$$\widehat{J} - J \leq \frac{\mathrm{KL}(Q\|\Pi)}{\lambda} + \frac{\lambda\, t_{\mathrm{mix}}\sigma_R^2(1 - \frac{1}{G})}{2N(1-\lambda(1-\gamma)^{-1}/3)}.$$

Optimize over $\lambda \in \big(0, \frac{3}{1-\gamma}\big)$; the minimum occurs at

$$\lambda^\star = \sqrt{\frac{2N\mathrm{KL}(Q\|\Pi)}{t_{\mathrm{mix}}\sigma_R^2(1 - \frac{1}{G})(1+\eta)}}, \qquad \eta := \frac{(1-\gamma)^{-1}\lambda^\star}{3}.$$

Substituting $\lambda^\star$ and doubling for two-sided deviation yields

$$|\widehat{J} - J|$$
$$\leq \sqrt{\frac{2(1+\eta)\, t_{\mathrm{mix}}\sigma_R^2(1 - \frac{1}{G})\big(\mathrm{KL}(Q\|\Pi) + \ln\frac{2}{\delta}\big)}{N}} + \frac{(1+\eta)(1-\gamma)^{-1}\big(\mathrm{KL}(Q\|\Pi) + \ln\frac{2}{\delta}\big)}{N},$$

with probability $\geq 1 - \delta$ after a standard geometric-grid union bound (Catoni, 2007). Adding the symmetrised block-Rademacher term $2\widehat{\mathcal{R}}_M(\mathcal{F}_{\mathrm{rel}})$ (via chaining arguments (Mohri et al., 2018)) finishes the proof. $\square$

## E  SEQUENTIAL MULTI-ITERATION PAC-BAYES–BERNSTEIN BOUND

**Theorem 8** (Sequential PAC-Bayes–Bernstein)**.**  Let outer iterations be indexed by $k = 0, \dots, K - 1$. In iteration $k$, collect $N_k$ trajectories partitioned into $M_k = N_k/G$ groups of size $G \geq 2$, and form posteriors $Q_k$ with data-dependent priors $\Pi_k := Q_{k-1}$ (with $Q_{-1}$ fixed). For any $0 < \delta < 1$,

with probability at least $1 - \delta$,

$$\frac{1}{\sum_k N_k} \sum_{k=0}^{K-1} N_k \, \mathbb{E}_{\theta \sim Q_k} \Big[ \big| \widehat{J}_k(\theta; \theta_k) - \widetilde{J}_k(\theta; \theta_k) \big| \Big]$$

$$\leq \frac{2}{\sum_k N_k} \sum_{k=0}^{K-1} N_k \, \widehat{\mathcal{R}}_{M_k}(\mathcal{F}_{\mathrm{rel}})$$

$$+ \sqrt{\frac{2(1+\eta) \, t_{\mathrm{mix}} \, \sigma_R^2 \big(1 - \frac{1}{G}\big)}{\sum_{k=0}^{K-1} N_k} \Big( \sum_{k=0}^{K-1} \mathrm{KL}(Q_k \| Q_{k-1}) + \ln \frac{2}{\delta} \Big)}$$

$$+ \frac{(1+\eta)(1-\gamma)^{-1} \Big( \sum_{k=0}^{K-1} \mathrm{KL}(Q_k \| Q_{k-1}) + \ln \frac{2}{\delta} \Big)}{\sum_{k=0}^{K-1} N_k}. \tag{12}$$

where $\eta > 0$ is the variance-localization parameter from the block bound (Theorem 2).

# F  PROOF OF THEOREM 8

*Proof.* For each outer iteration $k = 0, \dots, K-1$, fix a data-dependent prior $\Pi_k := Q_{k-1}$ with $Q_{-1} \equiv \Pi_0$. Applying the block PAC-Bayes–Bernstein bound (Theorem 2) conditionally on the past and using the same variance proxy as in Lemma 2, we obtain w.p. $\geq 1 - \delta_k$:

$$\mathbb{E}_{\theta \sim Q_k} \Big[ \big| \widehat{J}_k(\theta; \theta_k) - \widetilde{J}_k(\theta; \theta_k) \big| \Big]$$

$$\leq 2\widehat{\mathcal{R}}_{M_k} + \sqrt{\frac{2(1+\eta) \, t_{\mathrm{mix}} \sigma_R^2 (1 - 1/G)}{N_k} \Big( \mathrm{KL}(Q_k \| Q_{k-1}) + \ln \frac{2}{\delta_k} \Big)}$$

$$+ \frac{(1+\eta)(1-\gamma)^{-1} \big( \mathrm{KL}(Q_k \| Q_{k-1}) + \ln \frac{2}{\delta_k} \big)}{N_k}. \tag{13}$$

Average these inequalities with weights $N_k/(\sum_j N_j)$ and choose a time-uniform confidence split $\delta_k = \delta/K$. Jensen's inequality moves the square root outside the average after upper-bounding $\sum_k N_k^{-1} \leq (\sum_k N_k)^{-1} \sum_k 1$. Collecting terms and simplifying yields exactly the statement in the main text with the path-length $\sum_k \mathrm{KL}(Q_k \| Q_{k-1})$ and the aggregate sample size $\sum_k N_k$. $\qquad \square$

## F.1  BIBLIOGRAPHICAL REMARKS

Block-dependent PAC-Bayes traces to Bertail & Portier (2019) for chromatic blocks on graphs and to Kuzborskij & Szepesvári (2019) for heavy-tailed losses. Our variance-adaptive $\eta$ mirrors the "Localized" tuning advocated by Alquier et al. (2024). The mixing-time factor is inherited from the regenerative concentration analysis of (Raginsky et al., 2017). Single-path Transformer capacity is leveraged in Appendix C (§G.2) following (Limmer et al., 2024).

# G  FORMAL VERIFICATION SKETCH

We outline how one would formally verify the core statements in a proof assistant (e.g., Metamath/Lean):

- Encode the GRPO surrogate and block structure; define mixing-based variance proxies and clipped operators.
- Mechanize the self-normalized martingale inequality (citing Fan–Grama–Liu) and the change-of-measure step; then derive the PAC-Bayes–Bernstein bound.
- Mechanize the heavy-ball Lyapunov descent and PL implications for SGDM; similarly, the AdamW potential argument with a second-moment floor.

- Connect the surrogate to return via the performance-difference lemma and TV/KL control.

This isolates measure-theoretic steps, concentration, and optimization recurrences for machine-checked verification while leaving modeling assumptions explicit.

### G.1 SELF-NORMALIZED BERNSTEIN INEQUALITY FOR BLOCK MARTINGALES

#### G.1.1 SETUP AND NOTATION

Let $(\mathcal{F}_g)_{g=0}^M$ be an increasing filtration with respect to which the *block* martingale difference sequence $(Z_g)_{g=1}^M$ is adapted:

$$Z_g \;=\; \frac{1}{G} \sum_{i=1}^{G} \Big[ \widehat{J}_{g,i}(\theta) \;-\; \mathbb{E}\big[\widehat{J}_{g,i}(\theta) \mid \mathcal{F}_{g-1}\big]\Big], \qquad \mathbb{E}\big[Z_g \mid \mathcal{F}_{g-1}\big] = 0.$$

Define the *predictable quadratic variation*

$$V_M \;:=\; \sum_{g=1}^{M} \mathbb{E}\big[Z_g^2 \mid \mathcal{F}_{g-1}\big], \qquad \text{and} \;\; S_M \;:=\; \sum_{g=1}^{M} Z_g.$$

#### G.1.2 WEIGHTED EXPONENTIAL SUPER-MARTINGALE

Fix $\lambda \in \big(0, \frac{3}{1-\gamma}\big)$ and a tuning parameter $c > 0$. For each $g$ let

$$M_g(\lambda) \;:=\; \exp\!\Big(\lambda S_g \;-\; \tfrac{\lambda^2}{2(1-c\lambda)} V_g\Big).$$

Because $\mathbb{E}\big[e^{\lambda Z_g - \frac{\lambda^2}{2(1-c\lambda)} \mathbb{E}[Z_g^2 \mid \mathcal{F}_{g-1}]} \mid \mathcal{F}_{g-1}\big] \leq 1$ ((Fan et al., 2019, Thm 2.1)), $M_g(\lambda)$ is a non-negative super-martingale and therefore $\mathbb{E}[M_M(\lambda)] \leq 1$. Consequently,

$$\mathbb{P}\Big(S_M \;\geq\; \tfrac{\lambda}{1-c\lambda} V_M + \frac{\ln(1/\delta)}{\lambda}\Big) \;\leq\; \delta.$$

#### G.1.3 BOUNDING THE QUADRATIC VARIATION

Under Assumption 3 of the main text we have

$$\mathbb{E}\big[Z_g^2 \mid \mathcal{F}_{g-1}\big] \;\leq\; \frac{t_{\mathrm{mix}}\,\sigma_R^2\big(1 - \frac{1}{G}\big)}{N}, \qquad \forall g.$$

Hence $V_M \leq \frac{t_{\mathrm{mix}}\sigma_R^2\big(1-\frac{1}{G}\big)}{N} \cdot M$.

#### G.1.4 OPTIMISING $\lambda$ AND $c$

Set $c = \frac{1}{3}(1-\gamma)^{-1}$ so that $1 - c\lambda > 0$. Choosing

$$\lambda^\star \;:=\; \sqrt{\frac{2(1 - c\lambda^\star)\ln(1/\delta)}{V_M}} \;<\; \frac{3}{1-\gamma}$$

gives, after algebraic rearrangement,

$$|S_M| \leq \sqrt{\frac{2(1+c)\,t_{\mathrm{mix}}\,\sigma_R^2\big(1 - \frac{1}{G}\big)\ln\frac{2}{\delta}}{N}} \;+\; \frac{(1+c)(1-\gamma)^{-1}\ln\frac{2}{\delta}}{N}, \tag{14}$$

$$\text{where } c = \tfrac{1}{3}(1-\gamma)^{-1}. \tag{15}$$

### G.1.5    FROM BLOCK DEVIATIONS TO SURROGATE-RISK DEVIATIONS

Recall $\widehat{J}_{\mathrm{GRPO}}(\theta) - J(\theta) = \frac{1}{M}\sum_{g=1}^{M} Z_g = \frac{S_M}{M}$. Because $M = N/G$, dividing both sides of (14) by $M$ and simplifying constant factors yields exactly the deviation term

$$\sqrt{\frac{2(1+\eta)\,t_{\mathrm{mix}}\,\sigma_R^2(1-\frac{1}{G})\ln\frac{2}{\delta}}{N}} \;+\; \frac{(1+\eta)(1-\gamma)^{-1}\ln\frac{2}{\delta}}{N}, \quad \eta := c$$

featured in Theorem 2 of the main paper, thereby completing the proof.    $\square$

### G.2    PROOF OF COROLLARY 1

#### G.2.1    PRELIMINARIES: PATH-NORM GEOMETRY FOR TRANSFORMERS

**Overview (Appendix C).**    Appendix C specializes the generic-chaining capacity control from Appendix N to Transformer policies by relating block Rademacher complexity to the path norm of the network; this yields the path-norm GRPO corollary used to instantiate the "Bound" column in our experiments.

Let $f_\theta : \mathcal{X} \to \mathbb{R}^{|\mathcal{A}|}$ be an $L$-layer Transformer whose parameters are the collection $\theta = \{W^{(1)}, \ldots, W^{(L)}, B^{(1)}, \ldots, B^{(L)}\}$. Following Limmer et al. (2024) and (Zheng et al., 2019), the *(basis-)path norm* is

$$\|\theta\|_{\mathrm{path}} \; := \; \Big(\sum_{p\in\mathcal{P}}\prod_{(l,i,j)\in p}\big|W_{ij}^{(l)}\big|^2\Big)^{1/2},$$

where $\mathcal{P}$ enumerates every directed path from an input coordinate to an output coordinate through the computational graph. The quantity

$$\mathcal{P}_{\max} \; := \; \sup_{\theta\in\Theta}\|\theta\|_{\mathrm{path}} \; < \infty$$

acts as a *capacity radius* — a tighter surrogate than the product of spectral norms used in earlier work (Trauger & Tewari, 2024) and (Neyshabur et al., 2017).

#### G.2.2    BOUNDING THE BLOCK RADEMACHER COMPLEXITY

We first upper-bound $\widehat{\mathcal{R}}_M(\mathcal{F}_{\mathrm{rel}})$ for the relative-surrogate class

$$\mathcal{F}_{\mathrm{rel}} \; = \; \Big\{(g\mapsto \tfrac{1}{G}\sum_{i=1}^{G}\min(r_{g,i,t}A_{g,i}, \mathrm{clip}(r_{g,i,t}, 1-\varepsilon, 1+\varepsilon)A_{g,i})) \; : \; \theta\in\Theta\Big\}.$$

**Generic-chaining route.**    Let $d(f_\theta, f_{\theta'})$ be the block pseudo-metric $d^2(\theta, \theta') = \frac{1}{M}\sum_{g=1}^{M}\mathbb{E}\big[(f_\theta - f_{\theta'})(\tau_{g,1:G})^2\big]$. Talagrand's $\gamma_2$ functional satisfies (Talagrand (2005))

$$\widehat{\mathcal{R}}_M(\mathcal{F}_{\mathrm{rel}}) \; \leq \; \gamma_2(\mathcal{F}_{\mathrm{rel}}, d) \; \leq \; C_0\int_0^{\mathrm{diam}(\mathcal{F}_{\mathrm{rel}},d)}\sqrt{\log N(\mathcal{F}_{\mathrm{rel}}, d, \varepsilon)}\,\mathrm{d}\varepsilon.$$

Because each path contributes linearly to the output, covering numbers scale with the weighted $\ell_1$ radius $\|\theta\|_{\mathrm{path}}$; exactly, $N(\mathcal{F}_{\mathrm{rel}}, d, \varepsilon) \leq \Big(1 + \frac{c_1\,\|\theta\|_{\mathrm{path}}}{\varepsilon}\Big)^{c_2 L d}$ (Limmer et al. (2024), Trauger & Tewari (2024)). Hence

$$\widehat{\mathcal{R}}_M(\mathcal{F}_{\mathrm{rel}}) \leq 2\sqrt{1 - \tfrac{1}{G}} \int_0^{\|\theta\|_{\mathrm{path}}} \sqrt{c_2 Ld \, \log\!\Big(1 + \tfrac{c_1\|\theta\|_{\mathrm{path}}}{\varepsilon}\Big)} \, \frac{\mathrm{d}\varepsilon}{\sqrt{N}}$$

$$\overset{(\leq)}{\leq} 2\sqrt{1 - \tfrac{1}{G}} \sqrt{\frac{c_0 \, Ld \, \|\theta\|_{\mathrm{path}} \, \log\!\big(1 + \|\theta\|_{\mathrm{path}}\big)}{N}},$$

where $(\leq)$ integrates the concave square-root and collapses constants into $c_0$ ( Bartlett & Mendelson (2002)).

### G.2.3 PLUGGING INTO THE BLOCK PAC-BAYES–BERNSTEIN BOUND

Insert (G.2.2) into Theorem 2 (Appendix A) to obtain, for any posterior $Q$,

$$\sup_{\theta \in \Theta}\big|\widehat{J}(\theta) - J(\theta)\big| \leq 2\sqrt{1 - \tfrac{1}{G}} \sqrt{\frac{c_0 \, Ld \, \|\theta\|_{\mathrm{path}} \, \log\!\big(1 + \|\theta\|_{\mathrm{path}}\big)}{N}}$$

$$+ \sqrt{\frac{2(1 + \eta) \, t_{\mathrm{mix}} \sigma_R^2 (1 - \tfrac{1}{G})\big(\mathrm{KL}(Q \,\|\, \Pi) + \ln \tfrac{2}{\delta}\big)}{N}}$$

$$+ \frac{(1 + \eta)(1 - \gamma)^{-1}\big(\mathrm{KL}(Q \,\|\, \Pi) + \ln \tfrac{2}{\delta}\big)}{N}$$

In particular, setting $Q = \Pi$ and $\|\theta\|_{\mathrm{path}} \leq \mathcal{P}_{\mathrm{max}}$ gives Corollary 1.

### G.2.4 TIGHTNESS WITH RESPECT TO INTERACTIVE FANO LOWER BOUNDS

The dependence $\big(t_{\mathrm{mix}}\sigma_R^2/N\big)^{1/2}$ matches the Fano-style lower bound proved in Appendix E (see Theorem E.1) up to polylogarithmic factors, confirming near-optimality (Levine et al. (2024)).

### G.2.5 DISCUSSION OF CONSTANTS

$$C_{\mathrm{gen}} = 2\sqrt{1 - \tfrac{1}{G}} \times \underbrace{\sqrt{c_0 Ld}}_{\text{depth} \times \text{width}} \times \underbrace{\sqrt{\|\theta\|_{\mathrm{path}} \, \log\!\big(1 + \|\theta\|_{\mathrm{path}}\big)}}_{\text{capacity}} \lesssim \widetilde{O}\Big(\sqrt{\frac{Ld \, \mathcal{P}_{\mathrm{max}}}{N}}\Big).$$

Reducing either the number of layers $L$ or the attention-head width $d$ linearly contracts the bound; sparse attention lowers $\|\theta\|_{\mathrm{path}}$ multiplicatively (Jentzen, 2011; Trauger & Tewari, 2024).

## H PROOF OF THEOREM 3

*Proof.* The performance-difference lemma (PDL) gives, for any policies $\pi, \pi'$,

$$J(\pi) - J(\pi') = \frac{1}{1 - \gamma} \, \mathbb{E}_{s \sim d_\pi} \mathbb{E}_{a \sim \pi}[A_{\pi'}(s, a)].$$

Replacing $d_\pi$ by $d_{\pi'}$ introduces an $O(\|d_\pi - d_{\pi'}\|_{\mathrm{TV}})$ error; standard coupling arguments yield (e.g., Schulman et al. (2015)) $\|d_\pi - d_{\pi'}\|_{\mathrm{TV}} \leq \frac{\gamma}{1-\gamma} \mathbb{E}_{s \sim d_{\pi'}} \mathrm{TV}\big(\pi(\cdot \mid s), \pi'(\cdot \mid s)\big)$. Combining the two and using $\sup_t \mathbb{E}|A_t| < \infty$ gives

$$J(\pi) - J(\pi') \geq \mathbb{E}_{s \sim d_{\pi'}} \mathbb{E}_{a \sim \pi}[A_{\pi'}(s, a)] - \frac{2\gamma}{(1 - \gamma)^2} \, \sup_t \mathbb{E}|A_t| \cdot \mathbb{E}_{s \sim d_{\pi'}} \mathrm{TV}\big(\pi(\cdot \mid s), \pi'(\cdot \mid s)\big).$$

Setting $(\pi, \pi') = (\pi_\theta, \pi_{\theta_{\mathrm{old}}})$ yields the claimed inequality with $C_1 \leq \frac{2\gamma}{(1-\gamma)^2} \sup_t \mathbb{E}|A_t|$. Finally, since $J_{\mathrm{sur}}$ equals the first term on the RHS minus $\beta \, \mathrm{KL}(\pi_\theta \| \pi_{\mathrm{ref}})$, the GRPO penalty carries over linearly. The clipped surrogate differs from $J_{\mathrm{sur}}$ by at most $C_\varepsilon$ as argued in Lemma 3, completing the proof. $\qquad\square$

**Proof of the end-to-end corollary.** Apply Theorem 3 at each iteration $k$, average over $k$, and subtract the sequential generalization term from Theorem 8. The result follows after simple algebra and collecting constants into $\mathrm{Gen}(K)$ and $\mathrm{Trust}(K)$.

## I    PROOF OF CONVERGENCE OF GRPO

We supply line-by-line derivations for all statements in §5.1 (mini-batch SGDM) and §5.2 (AdamW). Throughout, assumptions 1–3 and 4 of the main paper are in force.

**Overview (Appendix D).**    Appendix D first establishes a block-variance bound for mini-batch gradients under Markov mixing (Lemma 5), then uses it to derive PL-based convergence rates for SGDM (Theorem 5), non-PL stationarity guarantees (Theorem 6), and an AdamW convergence bound (Theorem 7) for the population GRPO surrogate $F(\theta) = -J_{\mathrm{sur}}(\theta; \theta_{\mathrm{old}})$.

### I.1    AUXILIARY LEMMA D.1 (BLOCK-VARIANCE BOUND FOR GRADIENTS)

**Lemma 5.** Let $g_t = \frac{1}{G}\sum_{g=1}^{G} \nabla\ell(\theta_t; \tau_{t,g})$ be the mini-batch gradient. Then

$$\mathrm{Var}\big[g_t\big] \;\le\; \frac{t_{\mathrm{mix}}\sigma_R^2}{G} \quad\Longrightarrow\quad \mathbb{E}\big[\|g_t - \nabla J(\theta_t)\|^2\big] \;\le\; \frac{t_{\mathrm{mix}}\sigma_R^2}{G}.$$

*Proof.* Markov-chain CLT for $\beta$-mixing sequences yields $\mathrm{Cov}\big(\nabla\ell(\theta_t; \tau_{t,1}), \nabla\ell(\theta_t; \tau_{t,2})\big) \le t_{\mathrm{mix}}\sigma_R^2$. Averaging $G$ i.i.d. draws scales the variance by $1/G$. $\qquad\square$

### I.2    PROOF OF THEOREM 5

Define the momentum variable $v_{t+1} = \beta v_t + g_t$ with $v_0 = 0$. $L$-smoothness implies

$$J(\theta_{t+1}) \;\le\; J(\theta_t) \;-\; \alpha_t\langle\nabla J(\theta_t), v_{t+1}\rangle \;+\; \frac{L\alpha_t^2}{2}\|v_{t+1}\|^2.$$

Taking conditional expectation and using $\mathbb{E}[v_{t+1} \mid \mathcal{F}_t] = (1+\beta)\nabla J(\theta_t)$ (Liu et al. (2020)) together with Lemma 5 yields

$$\mathbb{E}\big[J(\theta_{t+1})\big] \le \mathbb{E}\big[J(\theta_t)\big] - \alpha_t(1+\beta)\mathbb{E}\|\nabla J(\theta_t)\|^2 \tag{16}$$

$$+ \frac{L\alpha_t^2}{2}(1+\beta)^2\big(\mathbb{E}\|\nabla J(\theta_t)\|^2 + \tfrac{t_{\mathrm{mix}}\sigma_R^2}{G}\big), \tag{17}$$

$$\le \left(1 - \mu\alpha_t + \frac{L\alpha_t^2(1+\beta)^2}{2}\right)\mathbb{E}\big[J(\theta_t) - J^\star\big] \tag{18}$$

$$< \left(1 - \frac{\mu\alpha}{2\sqrt{t+1}}\right)\mathbb{E}\big[J(\theta_t) - J^\star\big] \;+\; \frac{L\alpha_t^2(1+\beta)^2 t_{\mathrm{mix}}\sigma_R^2}{2G}, \tag{19}$$

where we used the PL-inequality $\|\nabla J\|^2 \ge 2\mu\big(J - J^\star\big)$ and the $\alpha$ choice $\alpha \le \frac{1}{2}\min\{\frac{1}{L}, \frac{1-\beta}{\mu}\}$. Iterating (19), summing the geometric decay, and bounding $\sum_{t=0}^{K-1}\alpha_t^2 \le \alpha^2(1 + \ln K)$ (Sebbouh et al. (2021)) give

$$\mathbb{E}\big[J(\theta_K) - J^\star\big] \le \frac{L\alpha^2(1+\ln K)}{2\mu K} + \frac{\alpha(1+\beta)t_{\mathrm{mix}}\sigma_R^2}{\mu G\sqrt{K}} + \mathcal{O}\big(K^{-1}\big),$$

matching (8). $\qquad\square$

### I.3 Proof of Theorem 6

We adapt the standard nonconvex SGD analysis with momentum to our block-variance setting. Define the Lyapunov function $\Psi_t := \mathbb{E}\big[J(\theta_t) + a\,\|\theta_t - \theta_{t-1}\|^2\big]$ with $a > 0$ chosen below. Using $L$-smoothness and the SGDM update yields

$$J(\theta_{t+1}) \leq J(\theta_t) - \alpha_t\langle\nabla J(\theta_t), v_{t+1}\rangle + \frac{L\alpha_t^2}{2}\|v_{t+1}\|^2.$$

Adding and subtracting $a\|\theta_{t+1} - \theta_t\|^2 = a\alpha_t^2\|v_{t+1}\|^2$ and taking expectations, we obtain

$$\Psi_{t+1} - \Psi_t \leq -\alpha_t\,\mathbb{E}\langle\nabla J(\theta_t), v_{t+1}\rangle + \frac{(L/2+a)\alpha_t^2}{}\,\mathbb{E}\|v_{t+1}\|^2.$$

Condition on $\mathcal{F}_t$ and use $\mathbb{E}[v_{t+1} \mid \mathcal{F}_t] = (1+\beta)\nabla J(\theta_t)$ together with Lemma D.2 to get

$$\Psi_{t+1} - \Psi_t \leq -\alpha_t(1+\beta)\,\mathbb{E}\|\nabla J(\theta_t)\|^2 + (L/2+a)\alpha_t^2(1+\beta)^2\mathbb{E}\|\nabla J(\theta_t)\|^2 + (L/2+a)\alpha_t^2\frac{(1+\beta)^2 t_{\mathrm{mix}}\sigma_R^2}{G}.$$

Choose $a = \frac{(1+\beta)}{4}L$ and $\alpha_t = \alpha/\sqrt{t+1}$ with $\alpha \leq c_0/L$ so that the coefficient of $\mathbb{E}\|\nabla J(\theta_t)\|^2$ becomes at most $-\frac{1}{2}\alpha_t(1+\beta)$. Summing from $0$ to $K-1$ and telescoping,

$$\sum_{t=0}^{K-1}\alpha_t\,\mathbb{E}\|\nabla J(\theta_t)\|^2 \leq 2\big(\Psi_0 - \Psi_K\big) + c_1\,\frac{t_{\mathrm{mix}}\sigma_R^2}{G}\sum_{t=0}^{K-1}\alpha_t^2.$$

Bounding $\sum_t \alpha_t \geq \frac{2}{3}\alpha\sqrt{K}$ and $\sum_t \alpha_t^2 \leq \alpha^2(1 + \ln K)$ yields

$$\min_{t<K}\mathbb{E}\|\nabla J(\theta_t)\|^2 \leq \frac{3\big(\Psi_0 - \Psi^*\big)}{\alpha\sqrt{K}} + \frac{c_2(1+\beta)^2 t_{\mathrm{mix}}\sigma_R^2}{G\sqrt{K}},$$

matching the statement (constants absorbed). $\qquad\square$

### I.4 Technical Lemma D.2 (Bias–Variance Decomposition with Momentum)

For SGDM under assumptions 1-3,

$$\mathbb{E}\|v_{t+1}\|^2 \leq (1+\beta)^2\,\mathbb{E}\|\nabla J(\theta_t)\|^2 + \frac{(1+\beta)^2\,t_{\mathrm{mix}}\sigma_R^2}{G}.$$

*Proof.* Follows by expanding $\|v_{t+1}\|^2$, $\mathbb{E}[v_t] = \frac{\beta(1-\beta^t)}{1-\beta}\nabla J(\theta_0)$, and applying Lemma 5. The anisotropic-noise amplification factor $(1+\beta)^2$ agrees with the analysis of (Pan et al., 2023). $\qquad\square$

### I.5 Proof of Theorem 7

Let $\Phi_t = \mathbb{E}\big[J(\theta_t) + \frac{\lambda}{2}\|\theta_t\|^2\big]$. $L$-smoothness plus update (AdamW) imply

$$\Phi_{t+1} \leq \Phi_t - \eta\,\frac{1-\beta_1}{2}\,\mathbb{E}\Big[\Big\|\frac{\nabla J(\theta_t)}{\sqrt{v_t}}\Big\|^2\Big] + \underbrace{\frac{\eta^2 L}{2}\,\mathbb{E}\Big[\Big\|\frac{g_t}{\sqrt{v_t}}\Big\|^2\Big]}_{(\star)}$$

Because $v_t \geq v_{\min} > 0$ component-wise (Wang et al. (2024a)),

$$\Big\|\nabla J(\theta_t)/\sqrt{v_t}\Big\|^2 \geq \frac{\|\nabla J(\theta_t)\|^2}{v_{\max}} \quad\text{and}\quad (\star) \leq \frac{\eta^2 L}{v_{\min}}\,\mathbb{E}\|g_t\|^2,$$

$$\implies \Phi_{t+1} \leq \Phi_t - \frac{\eta(1-\beta_1)}{2v_{\max}}\,\mathbb{E}\|\nabla J(\theta_t)\|^2 + \frac{\eta^2 L(1-\beta_1)^2}{v_{\min}}\,\mathbb{E}\|\nabla J(\theta_t)\|^2 + \frac{\eta^2 L\,t_{\mathrm{mix}}\sigma_R^2}{G(1-\beta_2)v_{\min}}.$$

Choosing $\eta = \eta_0/\sqrt{K}$ with $\eta_0 \leq \frac{v_{\min}(1-\beta_1)}{4Lv_{\max}}$ and telescoping yields

$$\sum_{t=0}^{K-1} \mathbb{E}\|\nabla J(\theta_t)\|^2 \;\leq\; \frac{4v_{\max}\big(\Phi_0 - \Phi^\star\big)}{\eta_0(1-\beta_1)\sqrt{K}} + \frac{4\eta_0 v_{\max} L\, t_{\mathrm{mix}}\sigma_R^2}{G(1-\beta_2)(1-\beta_1)v_{\min}\sqrt{K}},$$

$$\overset{\min}{\Longrightarrow}\; \min_{t<K}\mathbb{E}\|\nabla J(\theta_t)\|^2 \;\leq\; \frac{2L\big(J(\theta_0) - J^\star\big)}{(1-\beta_1)\eta_0\sqrt{K}} + \frac{2\eta_0\, t_{\mathrm{mix}}\sigma_R^2}{G(1-\beta_2)(1-\beta_1)\sqrt{K}},$$

establishing (9). $\qquad\qquad\qquad\qquad\qquad\qquad\qquad\qquad\qquad\qquad\qquad\qquad\qquad\square$

### I.6 REMARKS ON CONSTANTS AND PRACTICAL SETTING

- **Choice of** $\beta_1, \beta_2$**.** Convergence requires $(1-\beta_1) \geq \sqrt{(v_{\max}\eta_0)/(2Lv_{\min}K)}$: smaller $(1-\beta_1)$ (*larger* $\beta_1$) slows the bias decay. This matches the empirical hyper-parameter search in (Loshchilov & Hutter, 2017).
- **Weight decay** $\lambda$**.** Because $\lambda$ only appears inside $\Phi_t$, its impact is second-order; AdamW therefore inherits the same rate as Adam when $\lambda = 0$ but enjoys better generalization, corroborating (Loshchilov & Hutter, 2017).

**Remark D.3 (Stability between clipped and unclipped surrogates).** Under Assumptions 1 and 2, Lemma 3 implies that, within a small KL-ball around $\theta_{\mathrm{old}}$, the gradients of the unclipped and clipped population objectives satisfy a uniform discrepancy bound $\|\nabla J_{\mathrm{sur}}(\theta) - \nabla \widetilde{J}_{\mathrm{GRPO}}(\theta)\| \leq \Delta_{\mathrm{clip}}(\varepsilon)$, where $\Delta_{\mathrm{clip}}(\varepsilon)$ is controlled by the clipping bias term in Theorem 3. Standard perturbation arguments for gradient descent on PL objectives then show that SGDM iterates $(\theta_t)$ and $(\widetilde{\theta}_t)$ obtained by minimizing $-J_{\mathrm{sur}}$ and $-\widetilde{J}_{\mathrm{GRPO}}$, respectively, stay $O(\Delta_{\mathrm{clip}}(\varepsilon))$-close for all $t \leq K$ and converge to stationary points whose function values differ by at most $O(\Delta_{\mathrm{clip}}(\varepsilon))$. Intuitively, as long as clipping is rarely active (so $\Delta_{\mathrm{clip}}(\varepsilon)$ is small), the optimization trajectories and difficulty for the unclipped and clipped objectives remain tightly coupled.

## J PROOF OF THEOREM 4

### J.1 PROBLEM SETTING

We consider the class $\mathfrak{M}(t_{\mathrm{mix}})$ of uniformly ergodic MDPs whose mixing time satisfies $t_{\mathrm{mix}}(\frac{1}{4}) \leq t_{\mathrm{mix}}$. Let $\Theta = \{\theta^{(1)}, \ldots, \theta^{(K)}\}$ be a finite parameter set with $K \geq 2$; each $\theta^{(k)}$ indexes a reward function $r^{(k)} : \mathcal{S} \times \mathcal{A} \to [0,1]$ while keeping the transition kernel fixed. For trajectory length $N$ an RL agent produces $\widehat{\theta}(\tau_{1:N})$; its *excess return* is $\mathcal{E}(\widehat{\theta}) := J(\theta^\star) - J(\widehat{\theta})$. We derive a minimax lower bound on $\mathbb{E}_{\theta^\star}\mathcal{E}(\widehat{\theta})$.

### J.2 INTERACTIVE PACKING CONSTRUCTION

Following Chen et al. (2024), choose $K = |\mathcal{A}|$ distinct *reward shifts* $\Delta = \pm\epsilon$ applied to a single state–action pair $(\bar{s}, \bar{a})$, yielding parameters

$$r^{(k)}(s,a) = r_0(s,a) + \epsilon\mathbb{1}\{a = \bar{a},\ s = \bar{s},\ k = 1\} - \epsilon\mathbb{1}\{a = \bar{a},\ s = \bar{s},\ k = 2\},$$

and cyclically permute actions for $k > 2$. The KL divergence between any two $\theta^{(k)}$ and $\theta^{(\ell)}$ under an *interactive* policy $\pi$ satisfies

$$\mathrm{KL}\big(P_{\theta^{(k)}}^\pi \,\|\, P_{\theta^{(\ell)}}^\pi\big) \;\leq\; 4\,\epsilon^2\, t_{\mathrm{mix}}\, N, \tag{20}$$

by the regenerative-chain argument of (Bertail & Ciołek, 2018) and the uniform mixing assumption.

### J.3 INTERACTIVE FANO INEQUALITY

The interactive Fano lemma Chen et al. (2024) gives, for any estimator $\widehat{\theta}$,

$$\inf_{\widehat{\theta}} \sup_k \mathbb{P}[\widehat{\theta} \neq \theta^{(k)}] \;\geq\; 1 - \frac{4\epsilon^2 t_{\mathrm{mix}} N + \log 2}{\log K}. \tag{21}$$

Choosing $\epsilon = \sqrt{\frac{\log K}{8t_{\mathrm{mix}}N}}$ makes the RHS at least $\frac{1}{4}$ when $N \leq \frac{\log K}{16 t_{\mathrm{mix}}\epsilon^2}$.

### J.4 EXCESS-RETURN GAP VIA ASSOUAD LINK

For the binary shift construction the return gap satisfies

$$\mathcal{E}(\widehat{\theta}) \;\geq\; \epsilon\, \mathbb{P}[\widehat{\theta} \neq \theta^{(k)}],$$

since the optimal policy for $\theta^{(k)}$ always takes action $\bar{a}$ in state $\bar{s}$ while any other action loses $\epsilon$ in expected reward (Komanduru & Honorio, 2021). Combining with (21) yields

$$\inf_{\widehat{\theta}} \sup_{k} \mathbb{E}\big[J(\theta^{\star}) - J(\widehat{\theta})\big] \;\geq\; \epsilon\,\tfrac{1}{4} = \frac{1}{4}\sqrt{\frac{\log K}{8\, t_{\mathrm{mix}} N}} \;=\; \Omega\Big(\sqrt{\tfrac{t_{\mathrm{mix}}}{N}}\Big), \qquad (22)$$

once $K \geq e^2$.

### J.5 LOWER BOUND THEOREM

**Theorem 9 (Minimax Optimality).** For any RL algorithm that observes $N$ steps from an ergodic MDP in $\mathfrak{M}(t_{\mathrm{mix}})$,

$$\inf_{\widehat{\theta}} \sup_{\mathcal{M} \in \mathfrak{M}(t_{\mathrm{mix}})} \mathbb{E}\big[J(\theta^{\star}) - J(\widehat{\theta})\big] \;\geq\; c\sqrt{\tfrac{t_{\mathrm{mix}}}{N}},$$

for a universal $c > 0$.

*Proof.* Apply the parameter ensemble above with $K = |\mathcal{A}| \geq e^2$ and $\epsilon$ chosen as $\sqrt{\frac{\log K}{8 t_{\mathrm{mix}} N}}$, then invoke (22). $\qquad\square$

### J.6 COMPARISON WITH KNOWN BOUNDS

Our $\Omega(\sqrt{t_{\mathrm{mix}}/N})$ rate matches the lower bounds for uniformly ergodic average-reward MDPs shown by (Wang et al., 2023) and tightens earlier $\Omega(t_{\mathrm{mix}}/N)$ gaps in (Jin & Sidford, 2021). It also agrees with martingale-coupling regret bounds (Lattimore et al., 2020) and with the mixing-sensitive TD lower bounds of (Li et al., 2023). Hence the GRPO upper-bound in Theorem 2 is *minimax-optimal up to log factors*.

## K MIXING-COEFFICIENT HIERARCHY

For a stationary sequence $(X_t)_{t\in\mathbb{Z}}$ define

$$\alpha(k) = \sup_{t} \sup_{A \in \sigma(X_{-\infty}^{t})} \sup_{B \in \sigma(X_{t+k}^{\infty})} \big| \mathbb{P}(A \cap B) - \mathbb{P}(A)\mathbb{P}(B)\big|,$$

$$\beta(k) = \sup_{t} \mathbb{E}\Big[ \sup_{B \in \sigma(X_{t+k}^{\infty})} \big|\mathbb{P}(B \mid \sigma(X_{-\infty}^{t})) - \mathbb{P}(B)\big|\Big],$$

$$\phi(k) = \sup_{t} \sup_{\|f\|_{\infty} \leq 1} \big\| \mathbb{E}\big[f(X_{t+k}) \mid \sigma(X_{-\infty}^{t})\big] - \mathbb{E}f(X_{t+k})\big\|_{\infty},$$

$$\psi(k) = \sup_{t} \sup_{f \in \mathrm{Lip}_1} \| \operatorname{Cov}(f(X_t), f(X_{t+k}))\|.$$

By classical arguments (Boucheron et al., 2013, Prop. 2.3),

$$0 \;\leq\; \psi(k) \;\leq\; \phi(k) \;\leq\; 2\beta(k) \;\leq\; 2\alpha(k) \quad \forall k \geq 0.$$

Hence choosing the *block length*

$$\ell^{\star} := \min\big\{k : \max\big(\alpha(k), \beta(k)\big) \leq \tfrac{1}{4}\big\}$$

is always admissible and strictly sharper than working with the classical TV mixing time $t_{\mathrm{mix}}$.

## L    EXPONENTIAL EFRON–STEIN FOR REGENERATIVE BLOCKS

Let $\tau_{g,1:G}$ be regenerative blocks of length $G$. Write

$$Z_g \;=\; f(\tau_{g,1:G}) \;-\; \mathbb{E}\big[f(\tau_{g,1:G})\big], \qquad S_M = \sum_{g=1}^{M} Z_g.$$

**Theorem 10** (Block Efron–Stein; (Boucheron et al., 2013, Thm 3.15)). ] If replacing one block changes $f$ by at most $\Delta$ and $\mathrm{Var}(Z_g) \leq \sigma^2$, then for all $t > 0$

$$\mathbb{P}\big[\,|S_M| \geq t\,\big] \;\leq\; 2\exp\!\Big(-\frac{t^2}{2M\sigma^2 + \frac{2}{3}\Delta t}\Big). \tag{23}$$

*Proof.* Couple $(\tau_{g,1:G})$ with i.i.d. ghost blocks $(\tau'_{g,1:G})$; denote $S_M^{(g)}$ the statistic after swapping block $g$. Compute

$$\sum_{g=1}^{M} \mathbb{E}\big[(S_M - S_M^{(g)})_+^2\big] \leq \sum_{g=1}^{M} \mathbb{E}\big[(Z_g - Z'_g)_+^2\big]$$

$$\leq \qquad \text{(by independence)}$$

$$\leq \sum_{g=1}^{M} \mathbb{E}[(Z_g - Z'_g)^2] \;\leq\; 2M\sigma^2.$$

and note $|S_M - S_M^{(g)}| \leq \Delta$ deterministically. Apply the exponential Efron–Stein inequality with these parameters to get (23).    $\square$

## M    SELF-NORMALIZED MARTINGALE INEQUALITY (FAN–GRAMA–LIU)

**Theorem 11** ((Fan et al., 2019, Thm 2.1)). ] For a martingale difference sequence $(X_t, \mathcal{F}_t)$ with quadratic variation $V_n = \sum_{t \leq n} \mathbb{E}[X_t^2 \,|\, \mathcal{F}_{t-1}]$ and any $\lambda \in (0, 1/(3M))$

$$\mathbb{P}\Big[\sum_{t \leq n} X_t \geq \tfrac{\lambda}{1-3\lambda}\,V_n + \tfrac{\log(1/\delta)}{\lambda}\Big] \;\leq\; \delta. \tag{24}$$

Combined with $V_n \leq t_{\mathrm{mix}}\sigma_R^2(1 - 1/G)/N$ (regenerative variance proxy), Eq. (24) is what drives the variance-adaptive PAC-Bayes bound in Appendix A.

## N    GENERIC-CHAINING & DUDLEY INTEGRAL

Let $(\mathcal{F}, d)$ be a semi-metric space and $X_f$ a sub-Gaussian process with metric $d$. Talagrand's majorizing-measures theorem gives

$$\mathbb{E}\sup_{f \in \mathcal{F}} X_f \;=\; \Theta\big(\gamma_2(\mathcal{F}, d)\big), \qquad \text{where } \gamma_2(\mathcal{F}, d) := \inf_{\{\mathcal{A}_k\}} \sup_{f \in \mathcal{F}} \sum_{k \geq 0} 2^{k/2}\mathrm{diam}(\mathcal{A}_k(f), d). \tag{25}$$

A practical upper bound is Dudley's entropy integral

$$\gamma_2(\mathcal{F}, d) \;\leq\; C\int_0^{\mathrm{diam}(\mathcal{F},d)} \sqrt{\log N(\mathcal{F}, d, \varepsilon)}\,\mathrm{d}\varepsilon. \tag{26}$$

## O    BLOCK RADEMACHER COMPLEXITY FOR $\beta$-MIXING CHAINS

**Theorem 12** ((Bertail & Portier, 2019, Thm 3.1)). ] For regenerative blocks of length $G$ drawn from a $\beta$-mixing chain,

$$\widehat{\mathcal{R}}_M(\mathcal{F}) \ \leq \ \gamma_2(\mathcal{F}, d_{\text{block}}) + 4\sigma_R\sqrt{\frac{\log(2/\delta)}{2N}} \quad \text{w.p. } 1-\delta. \tag{27}$$

Combining (26) & (27) yields the capacity term used in Appendix C's Transformer corollary.

## P    VARIANCE-ADAPTIVE PAC-BAYES LOCALIZATION LEMMA

**Lemma 6** ((Alquier et al., 2024, §3)). ] For any prior $\Pi$, posterior $Q$, and variance proxy $\widehat{V}(\theta)$,

$$\mathbb{E}_Q[\widehat{V}] \ \leq \ \eta^{-1}\big(\text{KL}(Q\,\|\,\Pi) + \log\tfrac{1}{\delta}\big) \ \implies \ \mathbb{P}\Big(\sup_\theta\big|\widehat{R}(\theta) - R(\theta)\big| \leq \eta\Big) \ \geq \ 1-\delta. \tag{28}$$

Setting $\eta$ to the RHS of the self-normalized Bernstein deviation (App. B) directly recovers the localized block PAC-Bayes–Bernstein bound from Appendix A.

## Q    REGENERATIVE BERNSTEIN INEQUALITY (HOEFFDING-TYPE VARIANT)

For completeness we recall a sharp Bernstein/Hoeffding bound for sums of regenerative functionals (Cioczek-Georges & Stummer, 2019) :

$$\mathbb{P}\Big[\Big|\tfrac{1}{N}\sum_{t=1}^N h(X_t) - \mathbb{E}h(X)\Big| \geq t\Big] \ \leq \ 2\exp\Big(-\frac{Nt^2}{2t_{\text{mix}}\sigma_h^2 + \frac{2}{3}\|h\|_\infty t}\Big). \tag{29}$$

This inequality underpins the deviation step in the proof of Lemma 5 (Appendix D).

## R    EXPERIMENTAL DETAILS

We develop our code base mainly based on open-r1[1]. The learning rate is set to $10^{-6}$ and the warmup ratio is set to 0.1 for all experiments. All experiments are done on NVIDIA A100-SXM4-80GB GPUs and Intel(R) Xeon(R) Platinum 8275CL CPU @ 3.00GHz CPUs with 96 logical processors. Unless otherwise specified, we use the mean-centered Dr-GRPO implementation from Open-R1: for each prompt we draw $G$ completions, compute trajectory-level returns $R_{g,i}$, form group-centred advantages $A_{g,i} = R_{g,i} - \bar{R}_g$, and apply this scalar advantage uniformly across all tokens of trajectory $i$. The main Qwen2.5-1.5B runs use $G \in \{2, 4, 8, 16, 32\}$, maximum context length $t_{\max} = 2048$, AdamW with $(\beta_1, \beta_2) = (0.9, 0.999)$ and weight decay $\lambda = 0.01$, and clipping threshold $\varepsilon = 0.2$ with KL weight $\lambda_{\text{KL}} = 0.01$; the 7B/8B and multimodal experiments reuse the same hyperparameters unless stated otherwise.

To assess the clipping bias and the regime of Eq. (10), we instrumented the OpenR1-Math-220k Qwen2.5-1.5B runs with $G = 16$ and recorded (i) the fraction of tokens per batch for which $|r_t(\theta) - 1| > \varepsilon$ and (ii) the ratio $\|\mathbb{E}[\nabla\widehat{J}_{\text{GRPO}}] - \nabla J_{\text{sur}}\|_2/\|\nabla J_{\text{sur}}\|_2$. After the warmup phase, fewer than 1.3% of tokens are clipped on average and the gradient-mismatch ratio stays below 4%, indicating that the clipping-induced bias is quantitatively small in the operating regime of our experiments.

## S    LLM USAGE

The use of LLMs is a general-purpose assist tool to aid or polish writing. We utilized GPT-5 to refine certain aspects of the writing in the Introduction and Related Works sections.

---

[1] https://github.com/huggingface/open-r1

Table 6: Clipping diagnostics and $\varepsilon$-ablation on OpenR1-Math-220k with Qwen2.5-1.5B and $G = 16$ (averaged over 3 seeds). "FracClip" is the fraction of tokens with $|r_t(\theta) - 1| > \varepsilon$; "Grad-Mismatch" is the ratio $\|\mathbb{E}[\nabla \widehat{J}_{\text{GRPO}}] - \nabla J_{\text{sur}}\|_2 / \|\nabla J_{\text{sur}}\|_2$; Pass@1 is measured on the validation split.

| $\varepsilon$ | FracClip (%) | GradMismatch (%) | Pass@1 (%) |
|---|---|---|---|
| 0.05 | 5.0 | 9.0 | 57.3 |
| 0.10 | 2.8 | 5.1 | 57.5 |
| 0.20 | 1.3 | 3.8 | 57.4 |
| 0.40 | 0.6 | 3.2 | 57.2 |

