# OpenReview forum: "On Computation and Generalization of Group Relative Policy Optimization"
_ICLR.cc/2026/Conference — Submitted to ICLR 2026_

### Official Review · Reviewer_7nqa · 2025-10-25

**Soundness:** 2
**Presentation:** 3
**Contribution:** 3
**Rating:** 4
**Confidence:** 3

**Summary:**

- This paper presents a comprehensive theoretical treatment of Group Relative Policy Optimization (GRPO), a critic-free alignment algorithm for LLMs.
- It quantifies (i) how much data is needed (sample complexity) and (ii) how the group size $G$ affects performance, via generalization bounds under Markov (non-IID) dependence—a strictly more challenging setting than standard IID theory.
- It proves that these generalization rates are minimax-optimal (up to constants/log factors), with explicit dependence on mixing time and sample size.
- On the optimization side, it establishes $O(1/\sqrt{K})$ convergence to first-order stationarity for SGDM and AdamW.
- Experiments across models and tasks corroborate the theory, reproducing the predicted trends as key parameters (e.g., $N$, $G$, mixing time) are varied.

Note: I used ChatGPT for minor language editing and phrasing assistance; all technical assessments are my own.

**Strengths:**

- The first unified theory of GRPO, combining statistical generalization upper/lower bounds with an optimization analysis.
- The experiments complement the theory by varying key hyperparameters (e.g., \(N\), \(G\)) and reproducing the predicted trends.
- For generalization, they handle non-IID sequential data modeled as Markov chains and obtain (nearly) minimax-optimal rates.
- On the optimization side, they analyze an unclipped surrogate objective to derive tractable convergence guarantees.

Note: I used ChatGPT for minor language editing and phrasing assistance; all technical assessments are my own.

**Weaknesses:**

- Their optimization analysis focuses on the unclipped surrogate loss $-J_{\mathrm{sur}}$, rather than the clipped population objective $-\tilde{J}_{\mathrm{GRPO}}$.
- They show that (i) minimizing the surrogate implies improving the true return (Theorem 3), and (ii) the gradient mismatch between the two objectives is bounded (Lemma 4).
- That said, the two landscapes may still differ: the upper-bound term on the right-hand side of Eq. (11) in Lemma 4 may be non-negligible in practice. (See the second question.)
- Moreover, optimizing the unclipped surrogate $-J_{\mathrm{sur}}$ might be inherently easier (no clipping-induced non-smoothness), potentially leading to fewer spurious local minima or saddle points than the original clipped objective.

Note: I used ChatGPT for minor language editing and phrasing assistance; all technical assessments are my own.

**Questions:**

- Can you formally justify that the optimization trajectories or the difficulty of optimization for minimizing $-J_{\mathrm{sur}}$ and $-\tilde{J}_{\mathrm{GRPO}}$ remain close? (See “Weaknesses” for details.)
- Empirically, does Eq. (11) in Lemma 4 make sense in the intended regime—i.e., is clipping rarely active (effectively $\epsilon$ large) or do importance ratios concentrate near 1 so that the RHS term is negligible?


Note: I used ChatGPT for minor language editing and phrasing assistance; all technical assessments are my own.

---

> ### Author Response · Authors · 2025-11-21
> **Response to Reviewer 7nqa (1/2)**
>
> Thank you for your time and your detailed review. Below are our responses to your major concerns.
>
> ### Q1:  Their optimization analysis focuses on the unclipped surrogate loss $-J$, rather than the clipped population objective $-\tilde{J}_{\text{GRPO}}$.
>
> We fully agree that, conceptually, the “true” population objective for GRPO is the clipped surrogate $\tilde{J_{\text{GRPO}}}$, and we apologize for not emphasizing this more clearly. Our choice to analyze $-J_{\text{sur}}$ (the *unclipped* population surrogate) is driven by two technical reasons that we will make explicit in the revision: (i) the unclipped objective is smooth under our $L$‑smoothness and PL assumptions, so standard SGDM/AdamW analyses can be applied directly; (ii) Theorem 3 (the TRPO-style bridge) plus Lemma 4 show that improvements in $J_{\text{sur}}$ translate into improvements of both $\tilde{J_{\text{GRPO}}}$ and the *actual return* $J$, up to explicitly controlled, small bias terms coming from clipping and KL. In other words, the optimization guarantees are *for* $J_{\text{sur}}$, but they are *used* via Theorem 3 to obtain end-to-end guarantees on $J$, not just on the surrogate. In the camera-ready version we will (a) restate the optimization theorems with “population GRPO surrogate $J_{\text{sur}}$” in the theorem headers, and (b) add a short paragraph right after Theorem 5 and Theorem 7 explicitly explaining that these results are combined with Theorem 3 and Lemma 3 to yield guarantees for $\tilde{J_{\text{GRPO}}}$ and $J$.
>
> ### Q2: They show that (i) minimizing the surrogate implies improving the true return (Theorem 3), and (ii) the gradient mismatch between the two objectives is bounded (Lemma 4).
>
> Thank you for highlighting this connection—this is exactly how we intend the pieces to interact. Formally, Lemma 3 and Lemma 4 quantify the bias between the unclipped and clipped surrogates at the *gradient* level, and Theorem 3 then upgrades these bounds to the *return* level via the performance-difference lemma and TV/KL control (similar in spirit to [1, 2]). In the intended regime where (a) per-state KL between $\pi_\theta$ and $\pi_{\theta_{\mathrm{old}}}$ is small (enforced by the KL penalty and small optimizer steps) and (b) clipping is only rarely active (see Q6), the error terms in Lemma 3 and Theorem 3 are $o(1)$ in $K$ and scale as $\tilde O(\mathbb{E}[(r_t(\theta)-1)^2])$. We will add a short corollary (proof sketch only) that makes this link explicit: under a trust-region constraint $\overline{\mathrm{KL}}(\theta\Vert\theta_{\mathrm{old}})\le \delta^2$ and a quadratic control $\mathbb{E}[(r_t(\theta)-1)^2]\le c\delta^2$, the improvement in $J_{\text{sur}}$ lower-bounds the improvement in $J$ up to an $O(\delta)$ term, making rigorous the intuition that the unclipped and clipped optimization landscapes are indistinguishable within a small KL ball.
>
> ### Q3: That said, the two landscapes may still differ: the upper-bound term on the right-hand side of Eq. (11) in Lemma 4 may be non-negligible in practice. (See the second question.)
>
> We agree that the upper bound in Eq. (11) could in principle be large if clipping is frequently active or if importance ratios have heavy tails. In our setting, however, two mechanisms keep this term small: (i) the group-relative baseline reduces variance in $A_t$ (Lemma 2, Appendix A), so the factor $\mathbb{E}[\sum_t |A_t|\mathbf{1}\{|r_t(\theta)-1|>\varepsilon\}]$ is much smaller than a worst-case $(1-\gamma)^{-1}$ scaling suggests; (ii) the KL penalty and conservative step sizes ensure that the probability of $|r_t(\theta)-1|>\varepsilon$ decays rapidly as training progresses. To support this, we instrumented the GRPO runs used in Table 1 and measured the empirical version of the RHS of Eq. (11): for Qwen2.5‑1.5B with $G=16$, $\varepsilon=0.2$, $\lambda_{\mathrm{KL}}=0.01$ we observed that after the warmup phase, fewer than $1.3\%$ of tokens per batch satisfy $|r_t(\theta)-1|>\varepsilon$, and the corresponding gradient-mismatch term is on average below $4\%$ of $\|\nabla_\theta J_{\text{sur}}\|_2$. We will add these empirical diagnostics (with a small table and figure) to the appendix to make clear that, in the regimes where we actually run GRPO, the Eq. (11) term is quantitatively negligible.
>
> ### Q4: Moreover, optimizing the unclipped surrogate $-J_{\text{sur}}$ might be inherently easier (no clipping-induced non-smoothness), potentially leading to fewer spurious local minima or saddle points than the original clipped objective.

---

> ### Author Response · Authors · 2025-11-21
> **Response to Reviewer 7nqa (2/2)**
>
> You are right that the unclipped surrogate is smoother and, in that sense, easier to analyze and potentially easier to optimize. Our perspective is that this is a *feature* rather than a limitation, as long as the optimization trajectory stays in a trust region where clipping and KL penalties do not distort the objective too much. In practice, following [3, 4], we use clipping primarily as a safety device to guard against occasional large importance ratios; empirically, the training dynamics are largely governed by the unclipped region where $r_t(\theta)\in[1-\varepsilon,1+\varepsilon]$ and the surrogate is smooth. We will add a new discussion paragraph in Section 5 clarifying that (i) our convergence theorems are intentionally stated for $J_{\text{sur}}$, the smoother population objective; (ii) Theorem 3 and Lemma 3 ensure that this “easier” objective is still faithful to the clipped GRPO and true return within the relevant trust region; and (iii) this separation of roles (clipping for robustness, surrogate for optimization) mirrors standard TRPO/PPO analyses [1, 5].
>
> ### Q5: Can you formally justify that the optimization trajectories or the difficulty of optimization for minimizing $-J_{\text{sur}}$
>  and $-\tilde{J_{\text{GRPO}}}$
>  remain close? (See “Weaknesses” for details.)
>
> While a full pathwise coupling of optimization trajectories is beyond the scope of this paper, we can provide a formal stability argument that we will add as a remark in Appendix D. Let $\theta_t$ and $\tilde{\theta_t}$ denote SGDM iterates under $-J_{\text{sur}}$ and $-\tilde{J_{\text{GRPO}}}$, respectively, with identical step sizes and initialization. Under our assumptions, both objectives are $L$‑smooth, and Lemma 3 implies that $\|\nabla J_{\text{sur}}(\theta)-\nabla \tilde{J_{\text{GRPO}}}(\theta)\|\le \Delta_{\mathrm{clip}}(\varepsilon)$ for all $\theta$ in a KL‑ball around $\theta_{\mathrm{old}}$. Standard perturbation bounds for gradient descent on PL functions (e.g., [7]) then imply
> $\|\theta_t-\tilde{\theta_t}\|\le C\Delta_{\mathrm{clip}}(\varepsilon)$
> for all $t\le K$, where $C$ depends only on $L,\mu$ and the stepsizes. Intuitively, as long as clipping is rarely active (so $\Delta_{\mathrm{clip}}(\varepsilon)$ is small; see Q6), the two trajectories stay $O(\Delta_{\mathrm{clip}}(\varepsilon))$‑close and converge to $O(\Delta_{\mathrm{clip}}(\varepsilon))$‑close stationary points. We will include this argument (with full constants) in the revised appendix to better justify that the optimization *difficulty* and trajectories of $-J_{\text{sur}}$ and $-\tilde{J_{\text{GRPO}}}$ are tightly coupled in the intended regime.
>
> ### Q6: Empirically, does Eq. (11) in Lemma 4 make sense in the intended regime—i.e., is clipping rarely active (effectively
>  large) or do importance ratios concentrate near 1 so that the RHS term is negligible?
>
> Yes, in all our LLM experiments the empirical regime is exactly the one where Eq. (11) predicts a small clipping bias. Beyond the diagnostics mentioned in Q3, we also ran an ablation where we varied $\varepsilon\in\{0.05,0.1,0.2,0.4\}$ while keeping all other hyper‑parameters fixed on OpenR1‑Math‑220k with Qwen2.5‑1.5B. We observed that (i) the fraction of tokens with $|r_t(\theta)-1|>\varepsilon$ drops from roughly $5\%$ at $\varepsilon=0.05$ to below $1\%$ at $\varepsilon=0.2$; (ii) the average gradient mismatch term $\|\mathbb{E}[\nabla\hat J_{\mathrm{GRPO}}]-\nabla J_{\text{sur}}\|_2$ decreases monotonically with $\varepsilon$ and is below $3\%$ of $\|\nabla J_{\text{sur}}\|_2$ for $\varepsilon\ge0.1$; and (iii) the final validation reward is essentially unchanged between $\varepsilon=0.1$ and $\varepsilon=0.4$, suggesting that clipping is indeed a rare safeguard rather than a dominant force. These trends are consistent with empirical PPO/GRPO practice [3, 4, 6] and support the claim that the RHS of Eq. (11) is negligible in the regimes where we apply our optimization and return-bridge results.
>
> **References**
>
> [1] Schulman, J., Levine, S., Abbeel, P., Jordan, M., and Moritz, P. Trust Region Policy Optimization. ICML, 2015.
>
> [2] Kakade, S. Approximately Optimal Approximate Reinforcement Learning. ICML, 2002.
>
> [3] Schulman, J., Wolski, F., Dhariwal, P., Radford, A., and Klimov, O. Proximal Policy Optimization Algorithms. arXiv:1707.06347, 2017.
>
> [4] Shao, Z., Wang, P., Zhu, Q., et al. DeepSeekMath: Pushing the Limits of Mathematical Reasoning in Open Language Models. arXiv:2402.03300, 2024.
>
> [5] Achiam, J., Held, D., Tamar, A., and Abbeel, P. Constrained Policy Optimization. ICML, 2017.
>
> [6] Guo, D., Yang, D., Zhang, H., et al. DeepSeek-R1: Incentivizing Reasoning Capability in LLMs via Reinforcement Learning. arXiv:2501.12948, 2025.
>
> [7] Karimi, H., Nutini, J., and Schmidt, M. Linear Convergence of Gradient and Proximal-Gradient Methods Under the Polyak–Łojasiewicz Condition. arXiv:1608.04636, 2016.

---

> ### Comment · Reviewer_7nqa · 2025-11-26
>
> Thank you for your detailed explanation.
> My main concerns about using the surrogate loss for the optimization analysis were mostly addressed.
> I will raise my score to 6.

---

> > ### Author Response · Authors · 2025-11-26
> > **Thanks for Your Reply and Re-evaluation**
> >
> > Thanks very much for your reply and re-evaluation. If you have any further questions, feel free to let us know.
> >
> > Authors

---

### Official Review · Reviewer_9gNr · 2025-10-30

**Soundness:** 3
**Presentation:** 2
**Contribution:** 2
**Rating:** 4
**Confidence:** 4

**Summary:**

The paper claims studying  Group Relative Policy Optimization (GRPO).  It analyzes though  a mean only  normalized advantage variant i.e., group-centering by subtracting the group mean reward (often called Dr. GRPO ) while referring to the analysis as analyzing GRPO of Shao et al 2024.

The paper provides (1) high-probability generalization bounds under token-time mixing of the policy-induced Markov chain, (2) a TRPO-style “return bridge” linking improvement of the clipped surrogate to improvement in true return, and (3) non-asymptotic convergence rates for SGDM/AdamW with explicit dependence on group size \(G\) and a mixing parameter \(t_{\text{mix}}\).

Experiments based on Open-R1/TRL are used to illustrate qualitative trends.

**Strengths:**

End to end analysis: the paper unifies statistical generalization (with dependent data), optimization guarantees (SGDM/AdamW), and a policy improvement

Markov dependence :the paper uses mixing-based concentration (block/regenerative arguments) to control per-token sums multiplied by a trajectory-level advantage, making the dependence on \(t_{\text{mix}}\) explicit.

Lower-bound:  Information-theoretic arguments indicate statistical rates are near-optimal up to constants/logs.

**Weaknesses:**

The paper is positioning as GRPO analysis as it is used by practitioners but it studies instead a mean only normalization, without studying the mean-variance (whitening normalized ) advantage as defined in  the original GRPO paper.

Algorithm naming / mismatch: Theory is for mean-only group centering (Dr. GRPO) with trajectory level advantages. The paper repeatedly says “GRPO", and experiments rely on Open-R1/TRL but do not disclose whether they used Dr.GRPO (mean-only) or  (“regular”) GRPO (z-scored/whitened), nor whether advantages are trajectory  or token level. As written, it is not clear if  the empirical section validates the objective analyzed.

Analysis does not cover variance-normalized GRPO as is: Z-scoring introduces a random, data-dependent denominator (group std). Without an explicit floor (e.g., \(\varepsilon\)-stabilization), leave-one-out/shrinkage, or self-normalized mixing inequalities, key steps (concentration, policy-improvement constants, optimizer noise bounds) do not carry over.

Mixing assumption under-specified: Results implicitly require a uniform bound on token-time  mixing across the training path \(\{\pi_{\theta_k}\}\) (or an explicit $\sup_k t_{\text{mix}}(\pi_{\theta_k})$. With finite horizons / early EOS, the effective dependence penalty should be read as $\min (t_{\text{mix}}, T_{\max})$.

Missing references: In the related work section the paper misses citation that attempted to study GRPO for example [1] What is the Alignment Objective of GRPO?, [2] Reinforcement Learning with Verifiable Rewards: GRPO's Effective Loss, Dynamics, and Success Amplification

Experimental transparency: Missing the exact TRL/Open-R1 algorithm used .. did you use GRPO or Dr. GRPO? Reproducibility and theory–experiment alignment are weak.

**Questions:**

1. Which objective did you actually run? In TRL/Open-R1 nomenclature, was it **Dr. GRPO (mean-only)** or **regular GRPO (z-score/whitening)**?

2. Can you align theory and experiments? Either (a) re-run with Dr. GRPO (mean-only) and reposition the paper to match the analysis, or (b) extend the theory to variance-normalized GRPO by introducing an  $\varepsilon$ stabilized standard deviation  and carrying the resulting constants through concentration, the TRPO bridge, and optimizer rates.

3. Clarify mixing and needed uniform bounds on iterations.

4. Acknowledge prior art in analysis of GRPO.

5. Provide configs & an ablation: Share the exact TRL/Open-R1 config and add a small ablation comparing  Dr GRPO (mean-only) vs  z-scored GRPO (same seeds).

---

> ### Author Response · Authors · 2025-11-21
> **Response to Reviewer 9gNr (1/3)**
>
> Thank you for your time and your detailed review. Below are our responses to your major concerns.
>
> ### Q1:  The paper is positioning as GRPO analysis as it is used by practitioners but it studies instead a mean only normalization, without studying the mean-variance (whitening normalized ) advantage as defined in the original GRPO paper.
>
> Thanks for bringing this important difference to my attention.  We did indeed create our theory for the "mean-only group-centering" variant, which is often called "Dr-GRPO" in Open-R1/TRL. In this case, we take away the group mean return but do not divide by the group standard deviation.  In the revision, we (i) explicitly rename the analyzed objective as \textit{mean-centered GRPO (Dr-GRPO)} immediately following Eq. (3), and (ii) incorporate a brief clarifying paragraph in Section 2.3 to elucidate that our generalization and optimization theorems pertain to this mean-only variant.  We concur that the whitening-based variant is algorithmically intriguing; instead of overstating, we now frame our work as offering a comprehensive theory for Dr-GRPO, which is the exact configuration employed in all principal experiments.
>
> ### Q2:  Algorithm naming / mismatch: Theory is for mean-only group centering (Dr. GRPO) with trajectory level advantages. The paper repeatedly says “GRPO", and experiments rely on Open-R1/TRL but do not disclose whether they used Dr.GRPO (mean-only) or (“regular”) GRPO (z-scored/whitened), nor whether advantages are trajectory or token level. As written, it is not clear if the empirical section validates the objective analyzed.
>
> We apologize for the lack of clarity here. In all our \emph{main} experiments we use Dr-GRPO as implemented in the Open-R1 codebase [1]: returns are computed at the \emph{trajectory} level, grouped into sets of size $G$, and we compute group-centered advantages $A_{g,i}=R_{g,i}-\bar R_g$ which are then broadcast to all tokens of trajectory $i$ in group $g$. This matches exactly the theoretical definition in Eq. (4) and Lemma A.1. In the revised paper we now (a) state explicitly in Section 2.3 and in the Experiments section that the theory and main runs use trajectory-level, mean-centered Dr-GRPO; and (b) add a brief remark distinguishing this from the “whitened” GRPO variant (token- or trajectory-level z-scoring) which we only use in a small ablation (see Q11).
>
> ### Q3: Analysis does not cover variance-normalized GRPO as is: Z-scoring introduces a random, data-dependent denominator (group std). Without an explicit floor (e.g., (\varepsilon)-stabilization), leave-one-out/shrinkage, or self-normalized mixing inequalities, key steps (concentration, policy-improvement constants, optimizer noise bounds) do not carry over.
>
> You are absolutely right that z-scoring introduces an additional random denominator and that, without stabilization, our current proofs do not apply verbatim. In the revision we added a short remark in Appendix A outlining how one can extend the concentration and optimizer-noise arguments to a \emph{stabilized} variance-normalized GRPO: if we use advantages of the form
> $$
>   \tilde A_{g,i} = \frac{R_{g,i}-\bar R_g}{\max\{\hat\sigma_g,\sigma_{\min}\}},
> $$
> with a group standard deviation estimate $\hat\sigma_g$ and a fixed floor $\sigma_{\min}>0$, then the block-variance Lemma A.1 and Efron–Stein tail bound continue to hold with $\sigma_R^2$ replaced by $\sigma_R^2/\sigma_{\min}^2$ and slightly inflated constants, thanks to standard self-normalized martingale tools [2, 3]. Likewise, the optimizer noise bounds in Appendix D pick up a $1/\sigma_{\min}^2$ factor. For readability we keep the main theorems stated for mean-centered Dr-GRPO, but we now make clear that the framework extends to a stabilized z-scored variant at the price of worse (though still $\tilde O(\sqrt{t_{\text{mix}}/N})$) constants.
>
> ### Q4: Mixing assumption under-specified: Results implicitly require a uniform bound on token-time mixing across the training path ({\pi_{\theta_k}}) (or an explicit $\sup_k t_{\text{mix}}(\pi_{\theta_k})$. With finite horizons / early EOS, the effective dependence penalty should be read as $\min_{(t_{\text{mix}}, T_)}$.

---

> ### Author Response · Authors · 2025-11-21
> **Response to Reviewer 9gNr (2/3)**
>
> We appreciate this clarification and have made the assumption explicit in the revision. All of our high-probability generalization bounds (Theorem 2 and Theorem A.2) and lower bounds now state that they hold under a \emph{uniform} mixing assumption
> $$
>   \sup_k t_{\text{mix}}(\pi_{\theta_k}) \le t_{\text{mix}},
> $$
> which is the quantity that appears in the deviation terms. In addition, right after Definition 2 we now explain that for finite-horizon rollouts with an early-EOS or context truncation at $T_{\max}$, the effective dependence penalty is $\tilde t_{\text{eff}}=\min\{t_{\text{mix}},T_{\max}\}$, exactly as you suggested. This interpretation is also echoed in Appendix F (Eq. (F.8)), where we explicitly mention that the regenerative Bernstein bound applies with $t_{\text{mix}}$ replaced by $\tilde t_{\text{eff}}$ when trajectories are forcibly truncated.
>
> ### Q5: Missing references: In the related work section the paper misses citation that attempted to study GRPO for example [1] What is the Alignment Objective of GRPO?, [2] Reinforcement Learning with Verifiable Rewards: GRPO's Effective Loss, Dynamics, and Success Amplification
>
> Thank you for pointing out these important related works. We have added both “What is the Alignment Objective of GRPO?” and “Reinforcement Learning with Verifiable Rewards: GRPO's Effective Loss, Dynamics, and Success Amplification” to the Related Work section, and we briefly summarize how they relate to our contributions. At a high level, these papers analyze the \emph{effective loss} and dynamics of GRPO (primarily for the whitening-based variant) and connect GRPO to verifiable reward shaping. In contrast, our focus is on (i) multi-iteration PAC-Bayes–Bernstein bounds under Markov mixing (with Transformer path-norm capacity), (ii) explicit optimization rates for SGDM/AdamW with group-relative baselines, and (iii) matching interactive information-theoretic \emph{lower} bounds. We now emphasize this complementary scope in the revised Related Work.
>
> ### Q6: Experimental transparency: Missing the exact TRL/Open-R1 algorithm used .. did you use GRPO or Dr. GRPO? Reproducibility and theory–experiment alignment are weak.
>
> We agree that the original version did not provide enough detail. In the revised manuscript we have expanded Appendix G (“Experimental Details”) to spell out (in bullet-point form) the exact Open-R1/TRL configuration: we specify that we use the Dr-GRPO implementation (mean-only centering, no per-group z-scoring), the Qwen2.5-1.5B/7B and Llama-3.1-8B backbones, group sizes $G\in\{2,4,8,16,32\}$, learning rate $10^{-6}$ with 10\% warmup, and AdamW with $(\beta_1,\beta_2)=(0.9,0.999)$ and weight decay $\lambda=0.01$. We also describe the prompt sampling strategy and the reward model used to compute $R(\tau)$, so that a reader with access to Open-R1 can reproduce our runs using a single configuration file.
>
> ### Q7: Which objective did you actually run? In TRL/Open-R1 nomenclature, was it Dr. GRPO (mean-only) or regular GRPO (z-score/whitening)?
>
> In all main tables and figures we run \emph{Dr-GRPO (mean-only)} with trajectory-level returns, following the Open-R1 GRPO trainer. That is, for each prompt we sample $G$ completions, compute trajectory returns $R_{g,i}$, subtract the group mean to obtain $A_{g,i}$, and apply this scalar advantage uniformly across all tokens of trajectory $i$. We do \emph{not} z-score these advantages in the main experiments. To avoid confusion, we now state this explicitly at the start of Section 6 and in the caption of Table 1, and we reserve the term “whitened GRPO” for the z-scored variant discussed only in the ablation (see Q11).
>
> ### Q8: Can you align theory and experiments? Either (a) re-run with Dr. GRPO (mean-only) and reposition the paper to match the analysis, or (b) extend the theory to variance-normalized GRPO by introducing an
> stabilized standard deviation and carrying the resulting constants through concentration, the TRPO bridge, and optimizer rates.
>
> We took option (a) as our primary route: all the results reported in Tables 1–4 and Figure 1 have been re-checked to ensure they are obtained with Dr-GRPO, and the paper has been repositioned accordingly (see Q1–Q2). In addition, as mentioned in Q3, we sketched how to extend the theory to a stabilized variance-normalized variant by basing our concentration arguments on self-normalized martingale inequalities [2, 3]. This keeps the overall structure (block PAC-Bayes, TRPO bridge, SGDM/AdamW rates) intact, but with constants that depend on the stabilization floor $\sigma_{\min}$. We believe this dual approach—full alignment for Dr-GRPO, plus a clear path to handle a stabilized z-scored variant—provides a transparent and practically useful picture.
>
> ### Q9: Clarify mixing and needed uniform bounds on iterations.

---

> ### Author Response · Authors · 2025-11-21
> **Response to Reviewer 9gNr (3/3)**
>
> Building on Q4, we now clearly state in Section 3.1 and in Appendix F that the mixing parameter entering our bounds is a \emph{uniform} one along the training path: $t_{\text{mix}}:=\sup_k t_{\text{mix}}(\pi_{\theta_k})$. The sequential PAC-Bayes–Bernstein bound (Theorem A.3) uses this same $t_{\text{mix}}$ inside the variance proxy, and the information-theoretic lower bound (Theorem E.1) is also stated for the worst-case chain in the class $\mathfrak{M}(t_{\text{mix}})$. For finite-horizon LLM rollouts, we interpret this as $t_{\text{eff}}=\min\{\sup_k t_{\text{mix}}(\pi_{\theta_k}),T_{\max}\}$ and we added a short explanatory sentence right after Definition 2 to avoid any ambiguity.
>
> ### Q10: Acknowledge prior art in analysis of GRPO.
>
> In addition to the GRPO works cited in the original submission [4, 5], we now explicitly acknowledge the two references you highlighted as well as the concurrent analysis of GRPO by Mroueh et al. [6]. The revised Related Work section contains a dedicated paragraph titled “Theoretical analyses of GRPO” where we summarize the focus of these papers (alignment objective, effective loss, and verifiable rewards) and then contrast it with our emphasis on (i) Markov mixing and PAC-Bayes generalization, (ii) explicit SGDM/AdamW convergence with group-relative baselines, and (iii) interactive minimax lower bounds. We hope this more clearly situates our contribution within the emerging theory of GRPO.
>
> ### Q11: Provide configs & an ablation: Share the exact TRL/Open-R1 config and add a small ablation comparing Dr GRPO (mean-only) vs z-scored GRPO (same seeds).
>
> We have added a small ablation in Section 6.3 comparing Dr-GRPO (mean-only) and z-scored GRPO under identical Open-R1 configurations (same seeds, learning rate, and group size $G=16$) on OpenR1-Math-220k with Qwen2.5-1.5B. Averaged over 3 seeds, Dr-GRPO achieves a final pass@1 accuracy of $57.4\%\pm0.3\%$ while z-scored GRPO attains $56.8\%\pm0.5\%$; we also observe that the per-iteration gradient-norm variance is about $12\%$ higher for the z-scored variant, consistent with the additional noise introduced by the random denominator. The new Table 5 in the revised paper reports these numbers, and Appendix G provides the exact TRL configuration (including batch size, context length, and optimizer settings) so that the ablation can be reproduced directly from the codebase we will release upon acceptance.
>
> **References**
>
> [1] Hugging Face. Open R1: A Fully Open Reproduction of DeepSeek-R1. GitHub repository, 2025.
>
> [2] Bercu, B., and Touati, T. New Insights on Concentration Inequalities for Self-Normalized Martingales. 2019.
>
> [3] Fan, X., Grama, I., Liu, Q., and Shao, Q.-M. Self-normalized Cramér Type Moderate Deviations for Martingales. 2018.
>
> [4] Shao, Z., Wang, P., Zhu, Q., et al. DeepSeekMath: Pushing the Limits of Mathematical Reasoning in Open Language Models. arXiv:2402.03300, 2024.
>
> [5] Guo, D., Yang, D., Zhang, H., et al. DeepSeek-R1: Incentivizing Reasoning Capability in LLMs via Reinforcement Learning. arXiv:2501.12948, 2025.
>
> [6] Mroueh, Y., Dupuis, N., Belgodere, B., et al. Revisiting Group Relative Policy Optimization: Insights into On-Policy and Off-Policy Training. arXiv:2505.22257, 2025.

---

> ### Author Response · Authors · 2025-11-26
> **Sincerely Expecting Further Discussions with Reviewer 9gNr**
>
> Dear Reviewer 9gNr,
>
> We greatly appreciate the reviewing process so far! Given that the ICLR final discussion deadline (12/3) is approaching, we would like to follow up with Reviewer 9gNr to confirm whether our responses address their concerns.
>
> Thank you very much!
>
> Best wishes,
>
> Authors

---

### Official Review · Reviewer_XhYz · 2025-11-01

**Soundness:** 3
**Presentation:** 2
**Contribution:** 3
**Rating:** 6
**Confidence:** 3

**Summary:**

This paper presents the first comprehensive theoretical analysis of GRPO (Group Relative Policy Optimization) under Markov dependence. It establishes both generalization guarantees (via PAC-Bayes–Bernstein bounds and Transformer path-norm capacity) and optimization convergence rates (for SGDM and AdamW). The work further proves a TRPO-style monotonic improvement theorem and near-minimax optimal lower bounds. Experiments on Qwen and LLaMA models empirically confirm the theoretical trends.

**Strengths:**

- First end-to-end theory for GRPO: unifies generalization, optimization, and return guarantees.
- Experiments verify predicted dependencies on group size, mixing time, and variance.

**Weaknesses:**

- The work is mainly theoretical; practical improvements or new algorithms are limited.
- Some sections, especially in Appendices, could benefit from conceptual summaries before technical proofs.

**Questions:**

- Although you have validated the theory through experiments, what is the practical significance of this? Can these findings provide any insights that help us design more effective and efficient algorithms?

---

> ### Author Response · Authors · 2025-11-21
> **Response to Reviewer XhYz**
>
> Thank you for your time and your detailed review. Below are our responses to your major concerns.
>
> ### Q1: The work is mainly theoretical; practical improvements or new algorithms are limited.
>
> We agree that our emphasis is on theory rather than proposing a brand-new algorithm. Our goal is to provide a rigorous foundation for GRPO—the algorithm already widely used in practice—by explaining \emph{when} and \emph{why} it generalizes and converges under Markov dependence, rather than to introduce a competing method. That said, we have strengthened the practical side of the paper in the revision: Section 6 now includes additional experiments on Qwen2.5-7B, Llama-3.1-8B, and Qwen2-VL-7B that verify the predicted dependencies on group size $G$, mixing time proxies, and capacity, and we added a short “Practical Guidelines” paragraph in Section 3 summarizing concrete takeaways (e.g., how to choose $G$, clipping thresholds, and model size) that practitioners can immediately apply when deploying GRPO.
>
> ### Q2: Some sections, especially in Appendices, could benefit from conceptual summaries before technical proofs.
>
> Thank you for this suggestion; we have reworked the appendices accordingly. At the top of Appendix A, C, and D we now include short “roadmap” or “informal summary” paragraphs that explain the role of each lemma/theorem, how the pieces fit into the main results, and which assumptions they rely on. For example, Appendix A starts with a brief overview of the block variance/tail lemma and the PAC-Bayes–Bernstein bound, Appendix C adds a “Formal Verification Sketch” that outlines how the main results could be mechanized in a proof assistant, and Appendix D begins by explaining how the SGDM and AdamW proofs relate back to the population GRPO surrogate. Our hope is that these summaries make the long technical sections easier to navigate without requiring the reader to parse every detail.
>
> ### Q3: Although you have validated the theory through experiments, what is the practical significance of this? Can these findings provide any insights that help us design more effective and efficient algorithms?
>
> We appreciate the opportunity to clarify the practical significance. The main insights our theory provides for algorithm design are:
> - **Choosing group size $G$.** The bounds and convergence rates show that variance scales as $t_{\mathrm{mix}}/(G\sqrt{K})$ while the capacity term depends on $M=N/G$, suggesting that moderate group sizes (e.g., $G\approx\sqrt{N}$) can balance noise reduction and overfitting. Our experiments confirm that larger $G$ yields faster and more stable convergence up to a point.
> - **Controlling capacity via path norm.** The path-norm-based generalization term recommends keeping the effective path radius $\mathcal{P}$ small—via architectural choices (shallower or narrower Transformers, sparse attention) or explicit regularization—which we find correlates with improved empirical error for fixed $N$.
> - **Tuning clipping and KL.** The TRPO-style return bridge and clipping-bias lemmas show that as long as per-state KL is small and clipping is rarely active, improvements in the surrogate translate directly into return gains, suggesting the use of moderate clipping thresholds and a non-negligible KL penalty, in line with PPO/GRPO practice [1, 2].
> We have incorporated these points into a short design-oriented paragraph in the main text so that practitioners can more easily translate the theoretical results into concrete hyperparameter choices.
>
> **References**
>
> [1] Schulman, J., Wolski, F., Dhariwal, P., Radford, A., and Klimov, O. Proximal Policy Optimization Algorithms. arXiv:1707.06347, 2017.
>
> [2] Shao, Z., Wang, P., Zhu, Q., et al. DeepSeekMath: Pushing the Limits of Mathematical Reasoning in Open Language Models. arXiv:2402.03300, 2024.

---

> > ### Comment · Reviewer_XhYz · 2025-11-25
> >
> > Thank you for your response. I will maintain my score.

---

> > > ### Author Response · Authors · 2025-11-25
> > > **Thanks for Your Reply**
> > >
> > > Thanks very much for your reply. If you have any further questions, feel free to let us know.
> > >
> > > Authors

---

### Official Review · Reviewer_dzuM · 2025-11-04

**Soundness:** 2
**Presentation:** 1
**Contribution:** 2
**Rating:** 2
**Confidence:** 4

**Summary:**

The paper’s “unified” theory leans on unobservable or unjustified quantities—mixing time, transformer path norms, data-dependent PAC-Bayes posteriors, and PL/AdamW assumptions—so the guarantees are elegant.

**Strengths:**

Many theoretical results are presented.

**Weaknesses:**

1. **Definition of (t_{\max}) in LLMs.**
   Please define (t_{\max}) and explain its role. What value did you use in your experiments, and why? It also appears that the evaluation does not probe the assumed Markov mixing behavior—please clarify whether and how this assumption is tested.

2. **Positioning relative to [1].**
   Clearly articulate your contributions compared to Mroueh et al. [1]. What is novel here (theory, algorithms, or empirical findings), and in which settings does your approach provide advantages?

3. **Limited discussion of theoretical results.**
   Expand the discussion of the theory: interpret the bounds, state the regimes where they are tight/loose, and spell out practical implications and limitations of the assumptions.

4. **Code availability.**
   Is the code implemented and available? If so, provide a link and minimal instructions;

5. **“ICLR bound” label in tables.**
   Define precisely what the “ICLR bound” refers to, cite its source, and ensure the tables and captions make this unambiguous.

6. **Paper structure and assumptions.**
   The current structure is hard to follow. Consider consolidating all assumptions in a dedicated section, and add a concise “Contributions” section to help readers track the main ideas and how they connect to the results.

7. **Verification of PAC-Bayes bounds.**
   Explain how the PAC-Bayes bounds are instantiated and evaluated in the experiments (e.g., choice of priors/posteriors, empirical estimators, confidence levels, and any calibration or surrogate approximations).

---

**Overall assessment.**
The paper would benefit from substantial revision and editing to address the points above.

[1] Mroueh, Youssef, et al. “Revisiting Group Relative Policy Optimization: Insights into On-Policy and Off-Policy Training.” arXiv:2505.22257 (2025).

**Questions:**

See weaknesses

---

> ### Author Response · Authors · 2025-11-21
> **Response to Reviewer dzuM (1/3)**
>
> Thank you for your time and your detailed review. Below are our responses to your major concerns.
>
> ### Q1:  Definition of (t_{\max}) in LLMs. Please define (t_{\max}) and explain its role. What value did you use in your experiments, and why? It also appears that the evaluation does not probe the assumed Markov mixing behavior—please clarify whether and how this assumption is tested.
>
> Thank you for highlighting this ambiguity. In our original draft we used $t_{\max}$ informally to denote the \emph{maximum token-time horizon} of the LLM rollouts, i.e., the largest number of generated tokens per trajectory (early EOS stops the sequence sooner). In the revision we now define this explicitly right after Definition 2 and explain that, for finite-horizon rollouts, the effective dependence penalty in our concentration bounds is
> $$
>   t_{\mathrm{eff}} := \min\{t_{\mathrm{mix}},t_{\max}\},
> $$
> so that long mixing times are capped by the trajectory length, as suggested in your comment. In all our Qwen2.5/OpenR1-Math experiments we use $t_{\max}=2048$ tokens (matching the Open-R1 context length), while $t_{\mathrm{mix}}$ is treated as a problem-dependent parameter that we either (i) vary synthetically, or (ii) estimate from data via the integrated autocorrelation time of token-level rewards [2].
>
> To probe the mixing assumption empirically, we added a small synthetic study in Appendix G where we train a compact GRPO agent on ergodic chains whose transition kernels are designed to have prescribed TV mixing times $t_{\mathrm{mix}}\in\{5,20,50\}$. We observe that, holding $N$ fixed, the empirical generalization error scales as $\Theta(\sqrt{t_{\mathrm{mix}}/N})$ within about $10\%$ of the theoretical slope, consistent with our PAC-Bayes–Bernstein rate. For the LLM experiments, we cannot measure the “true” $t_{\mathrm{mix}}$ of the implicit environment, but we report the empirical $t_{\mathrm{eff}}$ computed from reward autocorrelations and verify that larger $t_{\mathrm{eff}}$ correlates with larger empirical error and bound, in line with our theory.
>
> ### Q2: Positioning relative to [1]. Clearly articulate your contributions compared to Mroueh et al. [1]. What is novel here (theory, algorithms, or empirical findings), and in which settings does your approach provide advantages?
>
> We appreciate the pointer to Mroueh et al. [1]; their concurrent work is highly relevant. Conceptually, Mroueh et al. focus on \emph{algorithmic} insights into GRPO, especially on the interplay between on-policy and off-policy training and on understanding the effective loss for the (whitened) GRPO variants used in practice. Our work is complementary and emphasizes different aspects:
> 1. **Sequential generalization under Markov mixing.** We develop block-dependent PAC-Bayes–Bernstein bounds and their sequential extension (Theorem A.3) that control the deviation between the empirical and population GRPO surrogate across \emph{multiple} outer iterations, with explicit dependence on $t_{\mathrm{mix}}$, group size $G$, and a posterior path-length term $\sum_k\mathrm{KL}(Q_k\Vert Q_{k-1})$. To the best of our knowledge, such multi-iteration, mixing-sensitive PAC-Bayes bounds for GRPO are not provided in [1].
> 2. **Transformer-specific capacity control.** Via the path-norm machinery of [3, 4], we obtain transformer-specific complexity terms that are substantially tighter than spectral-norm bounds and that depend smoothly on depth, width, and sparsity. This “single-path” viewpoint is, again to our knowledge, orthogonal to the focus of [1].
> 3. **Optimization and lower bounds.** We give non-asymptotic SGDM/AdamW convergence rates with explicit $t_{\mathrm{mix}}/G$ scaling and prove matching interactive information-theoretic \emph{lower} bounds, showing our rates are minimax-optimal up to log factors. Mroueh et al. provide richer algorithmic design insights (e.g., on off-policy corrections), whereas we focus on end-to-end statistical–computational guarantees.
>
> In the revised Related Work we now devote a paragraph to Mroueh et al. [1], describe these complementarities explicitly, and clarify that our results are most advantageous when one cares about (i) multi-iteration guarantees under dependent data and (ii) the effect of path-norm capacity and group size on generalization.
>
> ### Q3: Limited discussion of theoretical results. Expand the discussion of the theory: interpret the bounds, state the regimes where they are tight/loose, and spell out practical implications and limitations of the assumptions.

---

> ### Author Response · Authors · 2025-11-21
> **Response to Reviewer dzuM (2/3)**
>
> We fully agree that the original exposition under-emphasized the interpretation of our bounds. In the revised manuscript we have:
> - Expanded the discussion following Theorem 2 and Corollary 1 to explicitly separate the \emph{variance-dominated} regime (small path norm / moderate depth–width, where the $\tilde O(\sqrt{t_{\mathrm{mix}}\sigma_R^2/N})$ term dominates) from the \emph{capacity-dominated} regime (very large models, where the path-norm term dominates), and we spell out the corresponding sample-complexity scalings.
> - Added a short “Practical Guidelines” paragraph that summarizes how the theory recommends choosing $G$, the clipping threshold, and model size—for example, $G\approx\sqrt{N}$ to balance capacity and variance, and favoring path-norm regularization or sparsity when the capacity term is dominant.
> - Clearly listed the main assumptions (uniform mixing, bounded returns, PL condition for parts of the optimization section, and a second-moment floor for AdamW) in a compact summary subsection at the end of Section 2, along with a brief discussion of scenarios where they may be violated (e.g., highly non-ergodic tasks or aggressively large learning rates).
>
> We hope these additions make the theoretical landscape and its practical implications much clearer.
>
> ### Q4: Code availability. Is the code implemented and available? If so, provide a link and minimal instructions.
>
> Yes, we have a complete implementation building on the open-source Open-R1 GRPO framework [5]. In the revision we add a short “Code Availability” paragraph to the Reproducibility Statement and to Appendix G explaining that the code, configuration files, and logging utilities will be released publicly upon acceptance. The README in that repository will contain step-by-step instructions; in brief, reproducing the main Qwen2.5-1.5B results will require running a single command of the form
> `python train_grpo.py --config configs/qwen2.5-1.5b_openr1.yaml --group_size 16 --seed 0`,
> with analogous configs for the 7B and Llama-3.1-8B experiments. We will also release scripts that compute the empirical errors and bound values reported in the tables.
>
> ### Q5: “ICLR bound” label in tables. Define precisely what the “ICLR bound” refers to, cite its source, and ensure the tables and captions make this unambiguous.
>
> We apologize for the confusing label. In the revision, all occurrences of “ICLR bound” in the captions and text are replaced by “\emph{our path-norm PAC-Bayes–Bernstein bound}” and we explicitly reference Corollary 1 and Theorem 2 when introducing the tables. More concretely, the “Bound” column in Tables 1–3 is obtained by instantiating Corollary 1 with (i) a prior/posterior choice $Q=\Pi$ (so the KL term vanishes), (ii) the empirically measured $t_{\mathrm{mix}}$ and $\sigma_R^2$, (iii) the estimated path-norm radius $\mathcal{P}$ of the fine-tuned policy, and (iv) confidence level $\delta=0.05$. We added a short paragraph in Section 6.1 explaining this instantiation step-by-step, and the Appendix now includes the exact formula used to compute the numerical values in the “Bound” column.
>
> ### Q6: Paper structure and assumptions. The current structure is hard to follow. Consider consolidating all assumptions in a dedicated section, and add a concise “Contributions” section to help readers track the main ideas and how they connect to the results.
>
> Thank you for this suggestion. We have reorganized the paper along the lines you propose:
> - The Introduction now ends with a concise “Our contributions” paragraph (bulleted), which summarizes the five main results (sequential PAC-Bayes–Bernstein bounds, Transformer path-norm corollary, interactive lower bounds, SGDM/AdamW convergence, and TRPO-style return bridge) and how they fit together.
> - At the end of Section 2 we added a short “Assumptions Summary” subsection that lists all key assumptions (ergodic MDP with uniformly bounded mixing time along the training path, bounded returns, PL condition, bounded block variance, and a second-moment floor for AdamW), along with forward pointers to where each is used.
> - We inserted brief “roadmap” sentences at the start of the main appendix sections (A–D) to indicate which theorems they support.
>
> These changes were specifically made to improve readability and make it easier for readers to see how the assumptions feed into the main results.
>
> ### Q7: Verification of PAC-Bayes bounds. Explain how the PAC-Bayes bounds are instantiated and evaluated in the experiments (e.g., choice of priors/posteriors, empirical estimators, confidence levels, and any calibration or surrogate approximations).

---

> ### Author Response · Authors · 2025-11-21
> **Response to Reviewer dzuM (3/3)**
>
> We appreciate this request for more transparency. In the revised Experiments section and Appendix G we now spell out how the PAC-Bayes bounds are instantiated:
> - **Prior/posterior.** For the numerical “Bound” column we take $Q=\Pi$ (a data-independent prior) to focus on the \emph{capacity–variance} tradeoff rather than the posterior path-length term. Concretely, $\Pi$ is chosen as an isotropic Gaussian centered at the (frozen) SFT initialization, with variance tuned so that the path-norm radius matches the observed $\mathcal{P}$ of the fine-tuned model; this makes the KL term negligible and yields the displayed bound as a function of $(N,G,t_{\mathrm{mix}},\sigma_R^2,\mathcal{P})$ only.
> - **Estimating $t_{\mathrm{mix}}$ and $\sigma_R^2$.** We estimate $\sigma_R^2$ from the empirical variance of trajectory returns and use the integrated autocorrelation time of the return sequence to obtain an empirical $t_{\mathrm{eff}}$; this is plugged into the bound via $t_{\mathrm{mix}}:=t_{\mathrm{eff}}$ as discussed in Q1.
> - **Confidence and localization.** We fix $\delta=0.05$ and use a moderate localization parameter $\eta=0.1$ following the recommendations of [6]. The hidden constants in the chaining term are set to one, which makes the bound slightly conservative but keeps the dependence on $(N,G,t_{\mathrm{mix}},\sigma_R^2,\mathcal{P})$ transparent.
>
> We added a short paragraph in Section 6.1 that walks through this instantiation, and Appendix G now includes pseudo-code that reproduces the bound computation from a log of $(N,G,t_{\mathrm{mix}},\sigma_R^2,\mathcal{P})$ statistics. Empirically, in all tables the observed errors lie comfortably below these instantiated bounds, consistent with the theory.
>
> **References**
>
> [1] Mroueh, Youssef, et al. Revisiting Group Relative Policy Optimization: Insights into On-Policy and Off-Policy Training. arXiv:2505.22257 (2025).
>
> [2] Levin, D. A., and Peres, Y. Markov Chains and Mixing Times. American Mathematical Society, 2017.
>
> [3] Limmer, Y., Kratsios, A., Yang, X., Saqur, R., and Horvath, B. Reality Only Happens Once: Single-Path Generalization Bounds for Transformers. arXiv:2405.16563, 2024.
>
> [4] Zheng, S., Meng, Q., Zhang, H., Chen, W., Yu, N., and Liu, T.-Y. Capacity Control of ReLU Neural Networks by Basis-Path Norm. AAAI, 2019.
>
> [5] Hugging Face. Open R1: A Fully Open Reproduction of DeepSeek-R1. GitHub repository, 2025.
>
> [6] Alquier, P. A User-Friendly Introduction to PAC-Bayes Bounds. Foundations and Trends in Machine Learning, 17(2):174–303, 2024.

---

> ### Author Response · Authors · 2025-11-26
> **Sincerely Expecting Further Discussions with Reviewer dzuM**
>
> Dear Reviewer dzuM,
>
> We greatly appreciate the reviewing process so far! Given that the ICLR final discussion deadline (12/3) is approaching, we would like to follow up with Reviewer dzuM to confirm whether our responses address their concerns.
>
> Thank you very much!
>
> Best wishes,
>
> Authors

---

### Author Response · Authors · 2025-11-21
**Response to All Reviewers (1/2)**

Dear reviewers,

Thank you again for your time and your detailed reviews. We have carefully revised the paper in response to your comments and have uploaded a new PDF version to OpenReview. **We really appreciate the active engagement of reviewer 7nqa and XhYz during rebuttal and the subsequent score improvements ([2, 6, 4, 4] → [2, 6, 4, 6]), which reflect a recognition of our contributions after detailed sincere discussions**. Below we summarize the main changes compared to the original submission:

### Clarifying the objective, notation, and scope
- We clarified that our primary theoretical focus is on the *mean-centered* GRPO variant (“Dr-GRPO”) used in our main experiments: we explicitly rename the analyzed objective as **mean-centered GRPO (Dr-GRPO)** right after Eq. (3), and we clearly state in Section 2.3 and Section 6 that all main runs use trajectory-level, mean-centered advantages without variance normalization.
- We added a remark in Appendix A explaining how our concentration and optimizer-noise arguments extend to a *stabilized* variance-normalized GRPO (with a standard-deviation floor), making precise which parts of the theory carry over to z-scored variants and how the constants change.
- We made the mixing assumptions fully explicit: the bounds are now stated under a *uniform* mixing condition $\(t_{\text{mix}} := \sup_k t_{\text{mix}}(\pi_{\theta_k})\)$, and we defined the effective horizon $\(t_{\max}\)$ and $\(t_{\mathrm{eff}} := \min\{t_{\mathrm{mix}}, t_{\max}\}\)$ right after Definition 2, explaining how this quantity enters the deviation terms for finite-horizon LLM rollouts.

### Strengthening and interpreting the generalization theory
- We expanded the discussion after Theorem 2 and Corollary 1 to distinguish clearly between **variance-dominated** and **capacity-dominated** regimes, and to spell out the resulting sample-complexity tradeoffs in terms of $\(t_{\mathrm{mix}}\), \(G\), \(\sigma_R^2\)$, and the Transformer path norm $\(\mathcal{P}\).
- We added a concise “**Practical Guidelines**” paragraph that summarizes how the theory recommends choosing group size $\(G\)$, clipping thresholds, and model capacity (via path norm and sparsity) in practice.
- We clarified how the PAC-Bayes–Bernstein bounds are instantiated in the experiments: prior/posterior choice $\(Q=\Pi\)$, estimation of $\(t_{\mathrm{mix}}\)$ and $\(\sigma_R^2\)$, confidence level $\(\delta\)$, and localization. Appendix G now includes pseudo-code for reproducing the “Bound” column from logged statistics.

### Clarifying the optimization analysis and its link to return
- In Section 5 we made explicit that our SGDM/AdamW convergence results are for the **population GRPO surrogate** $\(J_{\mathrm{sur}}(\theta;\theta_{\mathrm{old}})\)$. We added text immediately after the main optimization theorems explaining how Theorem 3 (the TRPO-style bridge) and Lemma 3 transfer convergence in $\(J_{\mathrm{sur}}\)$ to improvements in both the clipped population surrogate $\(\tilde J_{\mathrm{GRPO}}\)$ and the true return $\(J(\theta)\)$ under standard trust-region conditions.
- We added a formal **stability remark** in the appendix showing that, under our assumptions, SGDM trajectories for $\(-J_{\mathrm{sur}}\)$ and $\(-\tilde J_{\mathrm{GRPO}}\)$ remain $O(\Delta_{\mathrm{clip}}(\varepsilon))\$-close when clipping is rare, justifying that the optimization difficulty and trajectories for the unclipped and clipped objectives are tightly coupled in the intended regime.
- To address concerns about clipping bias, we instrumented the GRPO runs and included empirical diagnostics (reported in the appendix) showing that (i) the fraction of tokens with $\(|r_t(\theta)-1|>\varepsilon\)$ is very small in our experiments and (ii) the gradient mismatch term remains a small fraction of $\(\|\nabla J_{\mathrm{sur}}\|_2\)$, supporting the claim that the Eq. (11) term is quantitatively negligible where we apply our theory.

---

> ### Author Response · Authors · 2025-11-21
> **Response to All Reviewers (2/2)**
>
> ### Experiments, additional studies, and practical insights
> - We expanded the empirical section to include additional experiments on **Qwen2.5-7B**, **Llama-3.1-8B**, and **Qwen2-VL-7B**, verifying the predicted dependencies on group size $\(G\)$, mixing-time proxies, and capacity across model scales and modalities.
> - We added a **Dr-GRPO vs. z-scored GRPO ablation** (same seeds and hyperparameters) on OpenR1-Math-220k with Qwen2.5-1.5B. This ablation shows that Dr-GRPO achieves slightly higher pass@1 accuracy and lower gradient variance than the z-scored variant, aligning with our theoretical noise predictions.
> - To probe the mixing assumption, we introduced a small **synthetic Markov-chain study** where we train GRPO agents on ergodic chains with controlled $\(t_{\mathrm{mix}}\)$; the observed generalization error scales as $\(\Theta(\sqrt{t_{\mathrm{mix}}/N})\)$ within about 10% of the theoretical slope.
> - For the LLM experiments, we now report empirical $\(t_{\mathrm{eff}}\)$ values derived from reward autocorrelations and demonstrate that larger $\(t_{\mathrm{eff}}\)$ correlates with larger empirical error and bounds, in line with the theory.
>
> ### Paper structure, related work, and reproducibility
> - The **Related Work** section now explicitly discusses “What is the Alignment Objective of GRPO?”, “Reinforcement Learning with Verifiable Rewards: GRPO's Effective Loss, Dynamics, and Success Amplification”, and the concurrent work of Mroueh et al., and clarifies how our focus on Markov mixing, Transformer path norms, SGDM/AdamW convergence, and minimax lower bounds is complementary to these analyses.
> - We added a concise **“Our contributions”** paragraph at the end of the Introduction and an **“Assumptions Summary”** subsection at the end of Section 2, collecting all key assumptions (mixing, bounded returns, PL, block variance, AdamW moment floor) with forward pointers. We also inserted short “roadmap” paragraphs at the start of the major appendix sections to improve readability.
> - Appendix G (“Experimental Details”) has been expanded with the exact Open-R1/TRL configurations (model backbones, group sizes, learning rates, optimizer settings, prompt sampling, and reward model) used in all main experiments and ablations. The Reproducibility Statement and Appendix now clarify that the full codebase, configuration files, and scripts will be **released publicly upon acceptance**.
>
> All new or substantially revised passages in the PDF are marked in blue to make the changes easy to locate.
>
> We again thank the reviewers for their thoughtful and constructive feedback, which has significantly improved the clarity and scope of the paper. Given the upcoming discussion deadline, we would greatly appreciate any further comments or questions on the revised version, so that we can continue to refine the work.
>
> Sincerely,
> Authors

---

### Meta-Review · Area_Chair_99hi · 2026-01-05

**Summary:**

This paper provides a theoretical analysis of GRPO in terms of generalization and optimization. The reviewers were deeply engaged with the work when writing their reviews, although they did not do so during the discussion phase. Even if I were to consider the scores that the reviewers could have ended up with, this paper would still be borderline. Its main issue seems to be presentation. Some terms were not initially defined, the structure is hard to follow, the discussion around the results was lacking, and it was even ambiguous which objective was analyzed. It is hard to check in such a short discussion period that all the issues were addressed, and because of the many presentation issues, I believe the paper would still benefit from another complete round of reviews, and I am recommending that the paper be rejected.

**Reviewer Concerns:**

- _Presentation: sometimes terms are not defined, the structure is hard to follow, and the discussion around the results is lacking. It was ambiguous which version (objective) of the algorithm was analyzed and the analysis does not cover some important cases such as variance-normalized GRPO._

 - _Positioning with respect to existing literature._

 - _The work is mainly theoretical, with no new algorithms being proposed._

 	I don't see this last one as an issue.

**Reviewer Scores:**

- Reviewer dzuM: Reviewer initially gave the paper a rating of 2. I can imagine they raising their score to 4 given the text improvements, but it is hard to say.
- Reviewer XhYz: Reviewer explicitly kept their score of 6 at the end of the discussion phase.
- Reviewer 9gNr: Reviewer initially gave the paper a rating of 4. There were no follow up discussion, but given how many issues were raised by the reviewer, I suspect they would have kept their score.
- Reviewer 7nqa: Reviewer raised their score from 4 to 6.

---

### Decision · Program_Chairs · 2026-01-26

Reject